# Tropospheric products of the 2nd GOP European GNSS reprocessing (1996-2014)

*Jan Dousa, Pavel Vaclavovic, Michal Elias*

*NTIS - New Technologies for the Information Society, Geodetic Observatory Pecný, RIGTC*
*250 66 Zdiby, Czech Republic*

Correspondence to: J. Douša (jan.dousa@pecny.cz)

## Abstract

In this paper, we present results of the 2nd reprocessing of all data from 1996 to 2014 from all stations in the European GNSS permanent network as performed at the Geodetic Observatory Pecný (GOP). While the original goal of this research was to ultimately contribute to new realization of the European terrestrial reference system, we also aim to provide a new set of GNSS tropospheric parameter time series with possible applications to climate research. To achieve these goals, we improved a strategy to guarantee the continuity of these tropospheric parameters and we prepared several variants of troposphere modelling. We then assessed all solutions in terms of the repeatability of coordinates as an internal evaluation of applied models and strategies, and in terms of zenith tropospheric delays (ZTD) and horizontal gradients with those of ERA-Interim numerical weather model (NWM) reanalysis. When compared to the GOP Repro1 solution, the results of the GOP Repro2 yielded improvements of approximately 50% and 25% in the repeatability of the horizontal and vertical components, respectively, and of approximately 9% in tropospheric parameters. Vertical repeatability was reduced from 4.14 mm to 3.73 mm when using the VMF1 mapping function, a priori ZHD, and non-tidal atmospheric loading corrections from actual weather data. Raising the elevation cut-off angle from 3° to 7° and then to 10° increased RMS from coordinates' repeatability, which was then confirmed by independently comparing GNSS tropospheric parameters with the NWM reanalysis. The assessment of tropospheric horizontal gradients with respect to the ERA-Interim revealed a strong sensitivity of estimated gradients to the quality of GNSS antenna tracking performance. This impact was demonstrated at the Mallorca station, where gradients systematically grew up to 5 mm during the period between 2003 and 2008, before this behaviour disappeared when the antenna at the station was changed. The impact of processing variants on long-term ZTD trend estimates was assessed at 172 EUREF stations with time-series longer than 10 years, resulting in most significant impact, site-specific, due to the non-tidal atmospheric loading followed by the impact of changing elevation cut-off angle from 3° to 10°. The other processing strategy had very small or negligible impact on estimated trends.

**Keywords:** GPS, reprocessing, zenith tropospheric delay, tropospheric horizontal gradients, coordinate time series, reference frame

## 1 Introduction

The US Global Positioning System (GPS) became operational in 1995 as the first Global Navigation Satellite System (GNSS). Since that time, this technology has been transformed into a fundamental technique for positioning and navigation in everyday life. Hundreds of GPS permanent stations have been deployed for scientific purposes throughout Europe and the world, and the first stations have collected GPS data for approximately the last two decades. In 1994, a science-driven global network of continuously operating GPS stations was established by the International GNSS Service, IGS

(http://www.igs.org) of the International Association of Geodesy (IAG) to support the determination of precise GPS/GNSS orbits and, clocks and earth rotation parameters, which are necessary for obtaining high-accuracy GNSS analyses for scientific applications. A similar network, but regional in its scope, was also organized by the IAG Reference Frame Sub-Commission for Europe (EUREF) in 1996, which was called the EUREF Permanent Network (EPN), http://epncb.oma.be (Bruyninx et al. 2012). Although its primary purpose was to maintain the European Terrestrial Reference System (ETRS), the EPN also attempted to develop a pan-European infrastructure for scientific projects and co-operations (Ihde et al. 2014). Since 1996, the EPN has grown to include approximately 300 operating stations, which are regularly distributed throughout Europe and its surrounding areas. Today, EPN data are routinely analysed by 18 EUREF analysis centres.

Throughout the past two decades, GPS data analyses of both global and regional networks have been affected by various changes in processing strategy and updates of precise models and products, reference frames and software packages. To reduce discontinuities in products, particularly within coordinate time series, homogeneous reprocessing was initiated by the IGS and EUREF on a global and regional scale, respectively. To exploit the improvements in these IGS global products, the $2^{nd}$ European reprocessing was performed in 2015-2016, with the ultimate goal of providing a newly realized ETRS.

Currently, station coordinate parameter time series from reprocessed solutions are mainly used in the solid earth sciences as well as to maintain global and regional terrestrial reference systems. Additionally, from an analytical perspective, the long-term series of estimated parameters and their residuals are useful for assessing the performances of applied models and strategies over a given period. Moreover, tropospheric parameters derived from this GNSS reanalysis could be useful for climate research (Yuan et al., 1993), due to their high temporal resolution and unrivalled relative accuracy for sensing water vapour when compared to other techniques, such as radio sounding, water vapour radiometers, and radio occultation (Ning, 2012). In this context, the GNSS Zenith Tropospheric Delay (ZTD) represents a site-specific parameter characterizing the total signal path delay in the zenith due to both dry (hydrostatic) and wet contributions of the neutral atmosphere, the latter of which is known to be proportional to precipitable water (Bevis et al. 1994).

With the $2^{nd}$ EUREF reprocessing, the secondary goal of the GOP was to support the activity of Working Group 3 of the COST Action ES1206 (http://gnss4swec.knmi.nl), which addresses the evaluation of existing and future GNSS tropospheric products, and assesses their potential uses in climate research. For this purpose, GOP provided several solution variants, with a special focus on optimal tropospheric estimates, including VMF1 vs. GMF mapping functions, the use of different elevation cut-off angles, and estimates of tropospheric horizontal gradients using different time resolutions. Additionally, in order to enhance tropospheric outputs, we improved the processing strategy in a variety of ways compared to the GOP Repro1 solutions (Douša and Václavovic, 2012): 1) by combining tropospheric parameters in midnights and across GPS week breaks, 2) by checking weekly coordinates before their substitutions in order to estimate tropospheric parameters, and 3) by filtering out problematic stations by checking the consistency of daily coordinates. The results of this GOP reprocessing, including all available variants, were assessed using internal evaluations of applied models and strategy settings, and external validations with independent tropospheric parameters derived from numerical weather reanalyses.

The processing strategy used in the 2<sup>nd</sup> GOP reanalysis of the EUREF permanent network is described
in Section 2 and, new approach that is developed to guarantee a continuity of estimated tropospheric
parameters at midnights as well as between different GPS weeks is summarised in Section 3. The
relationship between mean tropospheric horizontal gradients and the quality of low-elevation GNSS
tracking is explained in Section 4. The results of internal and external evaluations of GOP solution
variants and processing models are presented in Section 4 and, the assessment of impacts of specific
variants on estimated ZTD trends in Section 5. The last section concludes our findings and suggests
avenues of future research.

## 2   GOP processing strategy and solution variants

The EUREF GOP analysis centre was established in 1997, and contributed to operational EUREF
analyses until 2013 by providing final, rapid, and near real-time solutions. Recently, GOP changed its
contributions to that of a long-term homogeneous reprocessing of all data from the EPN historical
archive. The GOP solution of the 1<sup>st</sup> EUREF reanalysis (Repro1) (Völksen, 2011) comprised the
processing of a sub-network of 70 EPN stations during the period of 1996-2008. In 2011, for the first
time, GOP reprocessed the entire EPN network (spanning a period of 1996-2010) in order to validate
the European reference frame and to provide the first homogeneous time series of tropospheric
parameters for all EPN stations (Douša and Václavovic, 2012).
In the 2<sup>nd</sup> EUREF reprocessing (Repro2), GOP analysed data obtained from the entire EPN network from
a period of 1996-2014 using the Bernese GNSS Software V5.2 (Dach et al., 2015). The GOP strategy
relies on a network approach utilizing double-difference observations. Only GPS data from the EPN
stations were included according to official validity intervals provided by the EPN central Bureau
(http://epncb.oma.be). Two products were derived from the reprocessing campaign in order to
contribute to a combination at the EUREF level performed by the coordinator of analysis centres and
the coordinator of troposphere products: 1) site coordinates and corresponding variance-covariance
information in daily and weekly SINEX files and 2) site tropospheric parameters in daily Tro-SINEX files.
This GOP processing was clustered into eight subnetworks (Figure 1~~Figure 1~~) and then stacked into
daily network solutions with pre-eliminated integer phase ambiguities when ensuring strong ties to
IGS08 reference frame. This strategy introduced state-of-the-art models (IERS Conventions, 2010) that
are recommended as standards for highly accurate GNSS analyses, particularly for the maintenance of
the reference frame. Additionally, the use of precise orbits obtained from the 2<sup>nd</sup> CODE global
reprocessing (Dach et al., 2014) guaranteed complete consistency between all models on both the
provider and user sides. Characteristics of this GOP data reprocessing strategy and their models are
summarized in Table 1~~Table 1~~. Additionally, seven processing variants were performed during the GOP
Repro2 analysis for studying selected models or settings: a) applying tropospheric mapping function
model GMF (Böhm et al., 2006a) vs. VMF1 (Böhm et al., 2006b), the latter based on actual weather
information, b) increasing the temporal resolution of tropospheric linear horizontal gradients in the
north and east directions, c) using different elevation cut-off angles, d) modelling atmospheric loading
effects, and e) modelling higher-order ionospheric effects. Table 2~~Table 2~~ summarizes the settings and
models of solution variants selected for generating coordinate and troposphere products, which are
supplemented with variant rationales.
Within the processing, we screened station coordinate repeatabilities from weekly combined solutions
and we identified any problematic station for which north/east/up residuals exceeded 15/15/30 mm
or RMS of north/east/up coordinate component exceeded values 10/10/20 mm. Such station was a
priori excluded from the tropospheric product for the corresponding day. There were other standard
control procedures within the processing when individual station could have been excluded, e.g. if a)
less than 60% of GNSS data available, b) code or phase data revealed poor quality, c) station metadata
were found inconsistent with data file header information (receiver, antenna and dome names,
antenna eccentricities) and, d) phase residuals were too large for all satellites in the processing period
indicating a problem with station. Tropospheric parameters were estimated practically without
constrains (a priori sigma greater than 1 m) thus parameter formal errors reflect relative uncertainties
of estimates. Usually, large errors indicate the lack of observations contributing to the parameter.
During the troposheric parameter evaluations, we applied filter for exceeding formal errors of
estimated parameters (ZTD sigma greater than 3 mm, normal cases stay below 1 mm).

## 3 Ensuring ZTD continuity at midnights

When site tropospheric parameter time series generated from the 2$^{nd}$ EUREF reprocessing are applied
to climate research, they should be free of artificial offsets in order to avoid misinterpretations (Bock
et al., 2014). However, GNSS processing is commonly performed on a daily basis according to adopted
standards for data and product dissemination. Thus far, EUREF analysis centres have provided
independent daily solutions, although precise IGS products are combined and distributed on a weekly
basis. Station coordinates are estimated on a daily basis and are later combined to form more stable
weekly solutions. According to the EUREF analysis centre guidelines
(http://www.epncb.oma.be/ documentation/guidelines/guidelines analysis centres.pdf), weekly
coordinates should be used to estimate tropospheric parameters on a daily basis, but there are no
requirements with which to guarantee the continuity of tropospheric parameters at midnights.
Additionally, there are also discontinuities on a weekly basis, as neither daily coordinates nor hourly
tropospheric parameters are combined across midnights between corresponding adjacent GPS weeks.
The impact of a 3-day combination was previously studied when assessing the tropospheric
parameters stemming from the 2$^{nd}$ IGS reprocessing campaign 2016 in the GOP-TropDB (Győri and
Douša, 2016).  We compared two global tropospheric products provided by the analysis centre CODE
(Centre of Orbit Determination in Europe) differing only in the procedure of combining tropospheric
parameters from the daily original solutions. The first product, COF, was based purely on a single-day
solution while the second product, COD, on a 3-day combination (Dach et al., 2014). A sub-daily
statistics were calculated by comparing 2-hour ZTD estimates from both products during 2013. There
were no significant biases observed, but mean standard deviation estimated from differences reached
0.8 mm in ZTD over a day, but almost 1.8 mm close to the day boundaries. Similarly, a dispersion
characterized by 1-sigma over all stations reached 0.5 mm for the former, but up to 1.2 mm for the
latter. Actual differences in ZTDs could even be ~~significantly~~ large~~rr reaching up to several millimetres~~
~~or more as the middle values of 2-hour ZTD estimates could have been compared only, i.e. at 1:00 UTC~~
~~and 23:00 UTC~~, because this case used approximations leading to smooth low-resolution values close
to the day boundaries.
During the 1$^{st}$ GOP reprocessing, there was no way to guarantee tropospheric parameter continuity at
midnight, as the troposphere was modelled by applying a piecewise constant model. In these cases,

tropospheric parameters with a temporal resolution of one hour were reported in the middle of the
hour, as was originally estimated. In the 2nd GOP reprocessing, using again hourly estimates, we applied
a piecewise linear model for the tropospheric parameters. The parameter continuities at midnights
were not guaranteed implicitly, but only by an explicit combination of parameters at daily boundaries.
For the combination procedure we used three consecutive days while the tropospheric product stems
from the middle day. The procedure is done again for three consecutive days shifted by one day. A
similar procedure, using the piecewise constant model, was applied for estimating weekly coordinates
which aimed to minimize remaining effects in consistency at transition of GPS weeks (at Saturday
midnight). The coordinates of the weekly solution corresponding to the middle day of a three-day
combination were fixed for the tropospheric parameter estimates. In the last step, we transformed
the piecewise linear model to the piecewise constant model expressed in the middle of each hourly
interval (HR:30), which was saved in the TRO-SINEX format to support the EUREF combination
procedure requiring such sampling. The original piecewise linear parameter model was thus lost and
to retain this information in the official product in the TRO-SINEX format, we additionally stored values
for full hours (HR:00). Figure 2Figure 2 summarizes four plots displaying tropospheric solutions with
discontinuities in the left panels (a), (c) and enforcing tropospheric continuities in the right panels (b,
d). While the upper plots (a), (b) display the piecewise constant model, bottom plots (c), (d) indicates
the solution representing the piecewise linear model. The GOP Repro1 implementation is thus
represented by Figure 2Figure 2(a) plot while the GOP Repro2 solution corresponds to Figure 2Figure
2(d) and, alternatively Figure 2Figure 2(b).

These theoretical concepts were practically tested using a limited data set in 1996 (Figure 3). The
panels in Figure 3 follow the organization of the theoretical plots shown in Figure 2Figure 2;
corresponding formal errors are also plotted along with estimated ZTDs. Discontinuities are visible in
the left-hand plots and are usually accompanied by increasing formal errors for parameters close to
data interval boundaries. As expected, discontinuities disappear in the right-hand plots. Although the
values between 23:30 and 00:30 on two adjacent days are not connected by a line in the top-right plot,
continuity was enforced for midnight parameters anyway, as seen in the bottom-right plot. Formal
errors also became smooth near day boundaries, thus characterizing the contribution of data from
both days and demonstrating that the concept behaves as expected in its practical implementation.

# 4   Quality of the observations and impact on tropospheric gradients

Recently, we have developed a new interactive web interface to conduct tropospheric parameter
comparisons in the GOP-TropDB (Győri and Douša, 2016), which is being prepared for the IGS
Tropospheric Working Group web (http://twg.igs.org/). Using the interface, we observed large
systematic tropospheric gradients during specific years at several EPN stations. Generally, from GNSS
data, we can only estimate total tropospheric horizontal gradients without being able to distinguish
between dry and wet contributions. The former is mostly due to horizontal asymmetry in atmospheric
pressure, and the latter is due to asymmetry in the water vapour content. The latter is thus more
variable in time and space than the former (Li et al., 2015). Regardless, mean gradients should be close
to zero, whereas dry gradients may tend to point slightly more to the equator, corresponding to
latitudinal changes in atmosphere thickness (Meindl et al., 2004). Similarly, orography-triggered
horizontal gradients can appear due to the presence of high mountain ranges in the vicinity of the
station (Morel et al., 2015). Such systematic effects can reach the maximum sub-millimetre level, while
a higher long-term gradient (i.e. that above 1 mm), is likely more indicative of issues with site

instrumentation, the environment, or modelling effects. Therefore, in order to clearly identify these systematic effects, we also compared our gradients with those calculated from the ERA-Interim.

It is beyond the scope of this paper to investigate in detail the correlation between tropospheric horizontal gradients and effects such as, for example, antenna tracking performance. However, we do observe a strong impact in the most extreme case identified when comparing gradients from the GNSS and the ERA-Interim for all EPN stations. Figure 4Figure 4 shows the monthly means of differences in the north and east tropospheric gradients from the MALL station (Mallorca, Spain). These differences increase from 0 mm up to -4 mm and 2 mm for the east and north gradients, respectively, within the period of 2003/06 - 2008/10. Such large monthly differences in GNSS and NWM gradients are not realistic, and were attributed to data processing when long-term increasing biases dropped down to zero on November 1, 2008, immediately after the antenna and receiver were changed at the station. During the same period, also yearly mean ZTD differences to ERA-Interim steadily changed from about 3 mm to about -12 mm and immediately dropping down to -2 mm in 2008 after the antenna change.

The EPN Central Bureau (http://epncb.oma.be), operating at the Royal Observatory of Belgium (ROB), provides a web service for monitoring GNSS data quality and includes monthly snapshots of the tracking characteristics of all stations. The sequence of plots displayed in Figure 5Figure 5, representing the interval of interest (2002, 2004, 2006 and 2008), reveals a slow but systematic and horizontally asymmetric degradation of the capability of the antenna to track low-elevation observations at the station. Therefore, we analysed days of the year (DoY) 302 and 306 (corresponding to October 28 and November 1, 2008) with the in-house G-Nut/Anubis software (Václavovic and Douša, 2016) and observed differences in the sky plots of these two days. The left-hand plot in Figure 6Figure 6 depicts the severe loss of dual-frequency observations up to a 25° elevation cut-off angle in the South-East direction (with an azimuth of 90°-180°), which cause the tropospheric linear gradient of approximately 5 mm to point in the opposite direction. Figure 10 also demonstrates that an increasing loss of second frequency observations appears to occur in the East (represented as black dots). The right-hand plot in this figure demonstrates that both of these effects fully disappeared after the antenna was replaced on October 30, 2008 (DoY 304), resulting in the appearance of normal sky plot characteristics and a GLONASS constellation with one satellite providing only single frequency observations (represented as black lines).

This situation demonstrates the high sensitivity of the estimated gradients on data asymmetry, particularly at low-elevation angles. The systematic behaviour of these monthly mean gradients, their variations from independent data and a profound progress over time, seem to be useful indicators of instrumentation-related issues at permanent GNSS stations. It is also considered that gradient parameters can be valuable method as a part of ZTD data screening procedure (Bock et al., 2016).

Although the station MALL represented an extreme case, biases at other stations were observed too, e.g. GOPE (1996-2002), TRAB (1999-2008), CREU (2000-2002), HERS (1999-2001), GAIA (2008-2014) and others. Site-specific, spatially or temporally correlated biases suggest different possible reasons such as site-instrumentation effects including the tracking quality and phase centre variation models, site-environment effects including multipath and seasonal variation (e.g. winter snow/ice coverage), edge-network effects when processing double-difference observations, spatially correlated effects in reference frame realization and possibly others. The problematic stations and periods mentioned

above were however still included in comparisons and trend analysis because of the lack of objective
criteria for their identification, which should be studied in future.

## 5    Assessment of reprocessing solutions

GOP variants and reprocessing models were assessed by a number of criteria, including those of the
internal evaluations of repeatability of station coordinates, residuals at reference stations, and the
external validation of ZTDs and tropospheric horizontal gradients with data from numerical weather
model (NWM) reanalyses.

## 5.1    Repeatability of station coordinates

We used coordinate repeatability to assess the quality of models applied in GNSS analysis. To be as
thorough as possible, we not only assessed all GOP Repro2 variants but also assessed two GOP Repro1
solutions in order to discern improvements within the new reanalyses. The two Repro1 solutions
differed in their used reference frames and PCV models: IGS05 and IGS08.
Table 3Table 3 summarizes mean coordinate repeatability in the north, east and up components of all
stations from their weekly combinations. All GOP Repro2 solution variants reached approximately 50%
and 25% of the lower mean RMS of coordinate repeatability when compared to the GOP Repro1/IGS08
solution in its horizontal and vertical components, respectively. These values represent even greater
improvements when compared to the GOP-Repro1/IGS05 solution. Comparing these two Repro1
solutions clearly demonstrates the beneficial impact of the new PCV models and reference frames. The
observed differences between Repro2 and Repro1 also indicate an overall improvement of the
processing software from V5.0 to V5.2, and the enhanced quality of global precise orbit and earth
orientation products.
Various GOP Repro2 solutions were also used to assess the selected models. Variants GO0 and GO1
differ in their mapping functions (GMF vs VMF1) used to project ZTDs into slant path delays. These
comparisons demonstrate that vertical component repeatability improved from 4.14 mm to 3.97 mm,
whereas horizontal component repeatability decreased slightly. By increasing the elevation cut-off
angle from 3° to 7° (GO2) and 10° (GO3), we observed a slight increase in RMS from repeatability of all
coordinates. This can be explained by the positive impact of low-elevation observations on the
decorrelation of height and tropospheric parameters, despite the fact that applied models (such as
elevation-dependent weighting, PCVs, multipath) are still not optimal for including observations at
very low elevation angles. On the other hand, it should be noted that the VMF1 mapping function is
particularly tuned to observations at 3° elevation angle which leads to biases at higher elevation angles,
Zus et al. (2015).
The GO4 solution represents an official GOP contribution to EUREF combined products. It is identical
to the variant GO1, but applies a non-tidal atmospheric loading. Steigenberger et al. (2009) discussed
the importance of applying non-tidal atmospheric loading corrections together with precise a priori
ZHD model. It has been concluded that using mean, or slowly varying, empirical pressure values for
estimating a priori ZHD instead of true pressure values results in a partial compensation of atmospheric
loading effects which is the case of GO1 solution. A positive 10% improvement in height repeatability
was observed for the GO4 solution. Our improvement was slightly lower than in a global scope
reported by Dach et al. (2011) with an improvement of 10-20% over all stations. As the effect depends
on selected stations, a slightly higher impact in a global scale might be attributed to the station
distribution, particularly differences in term of latitude and altitude.
No impact was observed from the higher-order ionospheric effects (GO4 vs. GO5) in term of coordinate
repeatability. As the effect is systematic within the regional network (Fritsche et al., 2005) and it was
mostly eliminated by using reference stations in the domains of interest. The combination of
tropospheric horizontal gradients from 6-h to 24-h time resolution (GO4 vs. GO6), using the piecewise
linear model, had a negligible impact on the repeatability of station coordinates too.

## 5.2   Reference frame - residuals at fiducial stations

The terrestrial reference frame (Altamimi et al., 2001) is a realization of a geocentric system of
coordinates used by space geodetic techniques. To avoid a degradation of GNSS products, differential
GNSS analysis methods require a proper referencing of the solution to the system applied in the
generation of precise GNSS orbit products. For this purpose, we often use the concept of fiducial
stations with precise coordinates well-known in the requested system. Such stations are used to define
the geodetic datum while their actual position can be re-adjusted by applying a condition minimizing
coordinate residuals. None station is able to guarantee a stable monumentation and unchanged
instrumentation during the whole reprocessing period. Thus a set of about 50 stations, with 100 and
more time periods for reference coordinates, was carefully prepared for datum definition in the GOP
reprocessing. An iterative procedure was applied then for every day by comparing a priori reference
coordinates with actually estimated ones and excluding fiducial station exceeding differences by 5, 5
and 15 mm in north, east and up components.
Figure 7Figure 7 shows the evolution of the number of actually used fiducial stations (represented as
red dots) from all configured fiducial sites (represented as black dots) after applying an iterative
procedure of validation on a daily basis. This reprocessing began with the use of 16-20 fiducial stations
in 1996, and this number increased to reach a maximum of over 50 during the period from 2003-2011.
After 2011, this number decreased, due to a common loss of reference stations available from the last
realization of the global terrestrial reference frame without changes in its instrumentation. In most
cases, only 2 or 3 stations were excluded from the total number, however, this number is lower for
some daily solutions, indicating the removal of even more stations. The lowest number of fiducial sites
(12) was identified on day 209 of the year 1999 while, but low numbers were, generally, observed at
the beginning of the reprocessing period, in 1996. We observed consistent mean RMS errors for
horizontal, vertical, and total residuals of 6.47, 10.22, and 12.25 mm and 4.83, 7.94, 9.35 mm for daily
and weekly solutions, respectively, which demonstrate the stability of the reference system in the
reprocessing. The seasonality in height coordinate estimates characterized by the RMS of residuals
from the reference frame realization is dominated by errors due to modelling of the troposphere. We
believe, the main contribution stems from the insufficiencies in modelling of wet tropospheric delay,
as the effect has the most pronounced seasonal signal within the GNSS data analysis. Additionally, the
estimated station ZTD parameters and height are difficult to de-correlate. In the next section, the
strong seasonal variation in comparing zenith total delays estimated from GNSS and NWM data is
clearly visible.

## 5.3   Zenith total delays

We compared all reprocessed tropospheric parameters with respect to independent data from the
ERA-Interim global reanalysis (Dee et al. 2011) provided by the European Centre for Medium-Range
Weather Forecasts (ECMWF) from 1969 to the present. For the period of 1996-2014, we calculated
tropospheric parameters (namely ZTD and tropospheric horizontal linear gradients) from the NWM for
all EPN stations using the GFZ (German Research Centre for Geosciences) ray-tracing software (Zus et
al., 2014). The comparison of tropospheric parameters was performed by applying the linear
interpolation of GNSS parameters to the original NWM 6-hour representation, using the GOP TropoDB
(Győri and Douša, 2016). For monthly statistics discussed in this section, we applied an iterative
procedure for outlier detection using the 3-sigma criteria calculated from the compared ZTD or
gradient differences.
Table 4Table 4 summarizes comparisons of GNSS ZTDs, and tropospheric horizontal gradients, from all
GOP processing variants with those obtained from the ERA-Interim. Mean biases and standard
deviations were first calculated for each stations and each month and then mean and standard
deviation of these values were computed, characterizing dispersions of all statistical values over the
ensemble of stations.
The results in the table indicate a mean ZTD bias -1.8 mm for all comparisons (GNSS – NWM) suggesting
ZTDs achieved from the NWM reanalysis are drier than those obtained from GNSS reprocessing. Similar
biases have been observed for all other European GNSS re-processing products during the period of
1996-2014 (Pacione et al., 2017). On the other hand, when processing the ERA-Interim using two
different software and methodologies within the GNSS4SWEC Benchmark campaign (Dousa et al.,
2016) during May and June of 2013 in Central Europe, and by their comparing to two GNSS reference
products based on different processing methods, we observed bias differences within ±0.4 mm in ZTD.
As neither GNSS nor NWM is able to sense the troposphere with an absolute accuracy better than the
bias that we observed, we cannot make any conclusion, but its independence of the GNSS software. A
mixture of common processing aspects such as ~~scope o~~scale of GNSS network, applied tropospheric
model, precise orbit product and others could still cause such a small biases in GNSS analysis at least.
Comparing the results of the official GOP Repro2 solution (GO4) to those of the legacy solution (GO0)
demonstrates an overall improvement of 9% in term of accuracy, which corresponds to a similar
comparison between the EUREF Repro1 and Repro2 products (Pacione et al., 2017). The improvement
is assumed to be even larger (indicated by the coordinate repeatability) since the comparison of
tropospheric parameters is ~~, as the quality of ZTD retrievals~~ limited by a lower quality of reference
products derived from NWM~~are generally lower for NWM compared to GNSS from various intra-/inter-~~
~~technique comparisons~~ data (Douša et al., 2016, Kačmařík et al., 2017, Bock and Nuret, 2009).
Comparing the GO1 and GO0 variants demonstrates that the VMF1 mapping function outperforms
GMF in term of standard deviation if the elevation cut-off angle of 3° is used. The change of mapping
function together with the use of more accurate a priori ZHD, resulted in the ZTD standard deviation
improving from 8.8 mm (GO0) to 8.3 mm (GO1). However, bias was slightly increased which could be
partly attributed to the use of mean pressure model for a priori ZHD calculation and compensating
part of the non-tidal atmospheric loading (see Section 5.1). Using non-tidal atmospheric loading
corrections along with precise modelling of a priori ZHD contributed to a small reduction of the bias
from -2.0 mm to -1.8 mm and, mainly, to the improvement by reducing this ZTD accuracy to 8.1 mm
(GO4). This corresponds with the previous assessment of the repeatability of station coordinates.
Degradation in ZTD precision was also observed when the elevation cut-off angle was raised from 3°
to 7° (GO2) or 10° (GO3). No impacts on ZTD were, however, visible neither from additional modelling
of high-order ionospheric effects (GO5) nor from stacking of 6-hour horizontal gradients into daily
piecewise linear estimates (GO6).
Figure 8Figure 8 displays the time series of statistics from comparisons of the GOP official ZTD product
(GO4) with respect to the results of the ERA-Interim reanalysis. Mean bias and standard deviation were
derived from the monthly statistics of the 6-hourly GNSS-ERA differences. A 1-sigma range of the mean
values, represented by error bars, are additionally derived from all stations on a monthly basis.
Although the time series show homogeneous results over the given time span, a small increase in the
mean standard deviation over time likely corresponds with increasing number of EPN sites, rising from
approximately 30 to 300. The early years (1996-2001) also display a worse overall agreement in 1-
sigma range of mean values over all stations, which can be attributed to the varying quality of historical
observations and precise orbit products. The mean bias varies from −3 to 1 mm during the period of
1996-2014, with a long-term mean of -1.8 mm (Table 4Table 4). The long-term mean is also relatively
small compared to the ZTD mean 1-sigma range of 3-5 mm.

## 5.4   Tropospheric horizontal linear gradients

Additional GNSS signal delay due to the tropospheric gradients were developed by McMillan (1995).
The complete tropospheric model for the line-of-sight delay (ΔD$_T$) using parameters zenith hydrostatic
delay (ZHD), zenith wet delay (ZWD) and first-order horizontal tropospheric gradients G$_N$ and G$_E$, all
expressed in units of length, is described as follows

$$\Delta D_T = mf_h(e)ZHD + mf_w(e)ZWD + mf_g(e)\cot(e)[G_N\cos(A) + G_E\sin(A)]  \qquad (1)$$

where $e$ and $a$ are observation elevation and azimuth angles and $mf_h$, $mf_w$, $mf_g$ are hydrostatic, wet
and gradient mapping functions representing the projection from an elevation to the zenith. Horizontal
gradients should optimally represent a ZTD change in a distances for north and east directions as it
could be represented by terms $G_N\cot(e)$ and $G_E\cot(e)$ in the equation. However, the gradients need
to be parametrized practically with respect to observation elevation angle instead of the distance
theoretically applicable to the tropospheric effect at various elevation angles. The interpretation of
the tropospheric horizontal gradients in the Bernese software represents north and east components
of angle applied for the tilting the zenith direction in the mapping function with gradients representing
(in unit of length) the tilting angle multiplied by the delay in zenith (Meindl et al., 2004).
Similarly as in case of ZTD and coordinate assessment, Table 4 shows that tropospheric gradients
became worse when raising the elevation cut-off angle from 3° to 7° (GO2) or 10° (GO3). Mean
standard deviations of the GO2 and GO3 solutions increased by 8% and 12%, respectively, which is
valid for the whole period of monthly time series (not showedshown). No significant differences in
temporal variations of mean biases of the north and east tropospheric gradients variants were
identified while they shared a higher variability during the years 1996-2001. No impact of modelling of
high-order ionospheric effects (GO5) was observed. Statistics of GO4 and GO6 solutions compared to
ERA-Interim revealed that standard deviations dropped from 0.38 mm to 0.28 mm and from 0.40 mm
to 0.29 mm for the east and north gradients, respectively. Worse performance of the GO4 solution is
attributed to the fact that tropospheric horizontal gradients were estimated with a 6-h sampling
interval using the piecewise linear model with applying practically no absolute or relative constraints.
In such cases, increased correlations of the gradients with other parameters can cause instabilities in
processing certain stations at specific times; the gradients absorb some remaining errors in the GNSS

analysis model. The mean biases of the tropospheric gradients are considered to be negligible, but it was demonstrated in Section 4 that some large systematic effects were indeed discovered and attributed to the quality of GNSS signal tracking.

Figure 9Figure 9 displays monthly time series of statistics from comparisons of the GNSS and NWM tropospheric horizontal gradients in north and east directions. Two solutions are highlighted in order to demonstrate the impact of different parameter temporal resolutions; a 6-hour resolution is used for GO4 and a 24-hour resolution is used for GO6. Seasonal variations are mainly pronounced when observing mean standard deviations (top plot), whereas gradual improvement is more pronounced for mean biases (bottom plot). The reduction of the initial mean biases in horizontal gradients, and the corresponding 1-sigma ranges over the values from the ensemble of stations, can be attributed to the improved availability and quality of low elevation observation tracking. Elevation cut-off angles for collecting GNSS observations were initially configured station by station, ranging from 0° to 15°, until 2008 when the elevation cut-off angle 0° was recommended for all the stations.

Mean standard deviations and their 1-sigma ranges over all stations (Figure 9Figure 9, top plot) are lower by a factor of 1.3 for the solution with 24-hour resolution (GO6) compared to the 6-hour resolution (GO4); the impact is also pronounced especially in the early years of the dataset. The improvement factor ranges from 1.03 to 1.65 with the mean value of 1.35 overall stations and it is usually higher for years before 2001. Theoretically, with 4 times more observations in GO6 the standard deviation was expected to be divided by a factor of 2. This discrepancy indicates serialous correlations in errors which are among others stemming from the errors in precise products and models. Significant improvements, however, indicates possible correlations between tropospheric gradients and other estimated parameters, such as ambiguities, height and zenith total delays, and suggests a careful handling particularly when applying a sub-daily temporal resolution.

## 5.5 Spatial and temporal ZTD analysis

We performed spatial and temporal analyses of all processed variants in order to assess the impact of different settings on tropospheric products. Zenith tropospheric delays from all variants were compared in such a way to enable assessing impact of any single processing change: 1) GO1-GO0 for mapping function and more precise a priori ZHD model, 2) GO2-GO1 and GO3-GO1 for different elevation cut-off angle, 3) GO4-GO1 for non-tidal atmospheric corrections, 4) GO5-GO4 for higher-order ionospheric corrections and, 5) GO6-GO4 for temporal resolution tropospheric horizontal gradients. Station-specific behavior is out of this paper and will be studied in future.

Geographical maps of spatially distributed biases and standard deviations in ZTDs from all compared variants for the whole network are showed shown in Figure 10Figure 10 and Figure 11Figure 11. Additionally median, minimum and maximum values of station-wise total statistics are provided in Table 5Table 5. The comparisons demonstrated that the impact of the higher-order effect is fully negligible. Although overall mean biases in Table 5 are small, the GO1-GO0 comparison indicates a small negative bias over a majority of the stations, see Figure 10. Biases from the comparison of variants with different elevation cut-off angles strongly indicates a station-specific behavior with a positive bias for stations around Poland, which has not been explained yet. According to the table and Figure 11, the highest impact on standard deviations is found in the GO1 vs. GO0 solutions comparison. The effect is latitude dependent and it follows the increasing magnitude of ZTDs towards the equator.

~~Detailed study illustrated in~~ Figure 12~~Figure 12~~, Figure 13~~Figure 13~~ and Figure 14~~Figure 14 then~~
illustrate~~s~~ ZTD statistics with respect to the station latitude, ellipsoidal height and time, respectively.
Using VMF1 mapping function together with precise a priori ZHD from VMF1 instead of the GMF and
GPT models, respectively, see GO1 vs. GO0, we observe biases ranging from -1.52 to 0.70 mm and the
median value -0.36 mm and, according to Table 5~~Table 5~~, with a moderate latitudinal dependence, see
Figure 12~~Figure 12. A similar, but slightly larger negative bias of -0.94±0.28 mm, was reported Kacmarik~~
~~et al. (2017) studying 400 stations in the central Europe~~. Standard deviations range from 0.69 mm to
3.82 mm in Table 5~~Table 5~~ with a ~~profound~~ marked increase along with the latitude, Figure 12~~Figure~~
~~12~~, indicating the GPT performs worse at higher latitudes. This ~~fully corresponds to the results from~~ is
consistent with ~~the paper by~~ Steigenberger et al. (2009) demonstrating a partial compensation of the
atmospheric loading effect by using the GPT model. In case the atmospheric loading effect is not
corrected for, the errors are mostly assimilated to the zenith total delay parameters if station
coordinates are fixed on a weekly basis. Additionally, Figure 14~~Figure 14~~ shows the standard deviation
grows with time which might be explained by increased number of ~~can be attributed to the use of blind~~
~~(GMF) and actual-weather (VMF1) mapping functions. The mapping function affects an optimal use of~~
low-elevation observations with time~~, which were~~ ~~growing~~ in EUREF permanent network ~~with time~~ as
demonstrated for WTZR station in Figure 15~~Figure 15~~.
Biases obtained from the comparison of different elevation cut-off angles, i.e. variants 3°/7° (GO2-
GO1) and 3°/10° (GO3-GO1), range from -0.81 mm to 1.66 mm and -2.22 mm to 2.66 mm, respectively,
and standard deviations from 0.15 mm to 1.29 mm and 0.31 to 2.04 mm, see Table 5~~Table 5~~. Generally,
the impact of different elevation cut-off angle doesn't reveal any biases neither with respect to the
latitude (Figure 12~~Figure 12~~) nor the station height (Figure 13~~Figure 13~~). As expected, the impact is
larger for the GO3-GO1 differences and affected particularly some stations. Yearly biases exceeding
±2.5 mm were identified for BELL, DENT, MLVL, MOPS, POLV RAMO and SBG2 stations. Temporal
dependences in the GO2-GO1 and GO3-GO1 comparisons, Figure 14~~Figure 14~~, show that the scatter
of station-specific biases steadily grows in time which is assumed to be related to the higher availability
of low-elevation observations. On the other hand, a small impact is observed for the standard deviation
compared to the other studied effects. This indicates the elevation cut-off angle affects mainly ZTD
biases, which has been also reported by Ning and Elgered (2012).
Table 5~~Table 5~~ shows that biases due to the non-tidal atmospheric loading (GO4-GO1) range from -
2.29 mm to 5.55 mm, which is one of the largest impact compared to other comparison variants, and
standard deviations range from 0.68 mm to 4.72 mm that represents the second largest impact
compared to all other variants. Standard deviation larger than 3 mm was observed at some stations,
such as JOZE, MAD2, MADR, MDVO, MOPI, NYAL, SBG2, VENE and WETT. It should be emphasized this
comparison reflects differences due to the modelling of atmospheric loading corrections in GO4 and,
a partial compensation of the loading effect by zenith tropospheric delay estimates in the GO1 solution
variant. The differences are strongly station-dependent, but did not reveal any dependence on
latitude, see Figure 12~~Figure 12~~. It shows, however, some degradation in standard deviation during
the first years of the reprocessing, see Figure 14~~Figure 14~~. Since a similar degradation has not been
observed for other comparison variants, it can be related to the quality of pressure data used to
compute atmospheric loading.
The impact of higher-order ionospheric effect (GO5-GO4) is negligible at all stations demonstrating
total statistics for all stations within ±0.3 mm when applying the y-range about 10 times smaller than
in other panels of Figure 12Figure 12, Figure 13Figure 13 and Figure 14Figure 14. A strong latitudinal
dependence is, however, clearly visible in Figure 12Figure 12 as well as a temporal variability showing
yearly statisticpeaks up to ±0.4 mm, Figure 14Figure 14. Both dependences are due to the changing
magnitude of ionospheric corrections, generally increasing towards the equator and a daily noon, and
along with quasi periodic cycles of the solar magnetic activity, reaching peaks around years 2001 and
504 2014.

The impact of stacking tropospheric gradients from 6-hour to daily estimates (GO6-GO4) is almost
negligible in term of biases which stay below ±1 mm, Table 5Table 5 and Figure 10Figure 10. However,
standard deviations range from 0.76 mm to 2.46 mm and grow towards the equator, Figure 12Figure
12. That can be certainly attributed to the more difficult modelling of a local asymmetry in the
troposphere, which is generally increasing together with the increasing of the water vapor content.
There is no significant temporal variation observed in bias, in Figure 14, but a small decrease in
standard deviation (Figure 14). It can be attributed to a higher stability of the gradient estimates with
time, see Figure 9Figure 9, when supported with increased number of available low-elevation
observations.

# 6 Impact of variants on long-term ZTD trend estimates

We assessed the impact of solution variant on long-term ZTD trend estimates by analysing 172 EUREF
stations providing the time-series of data longer than 10 years. For each station, the trend analysis was
performed without any data homogenization or outlier rejection as our focus was only on assessing
the impact of solution variants on the trend estimates. The ZTD trends were estimated using the least
squares regression method applied on model (Weatherhead et al., 1998)

$$Y_t = \mu + \beta X_t + S_t + \varepsilon_t \tag{2}$$

where $\mu$ is the constant term of the model, $\beta X_t$ is the linear trend function with $\beta$ representing the
trend magnitude, $S_t$ represents the term modelled by the sine wave function of time $X_t$ including
annual, 2nd harmonics and daily variations, and finally $\varepsilon_t$ is the noise in the data.
Site-by-site estimated ZTD trends from all the variants are provided in supplementary materials
completed by time-span information, number of records and estimated mean formal errors calculated
over all variants. In total, trends range from -0.99 to 0.96 mm/year. Although the individual station
trend provided in supplements could be compared to other studies, e.g. Baldysz et al. (2016), Klos et
al. (2016) or Nilsson and Elgered (2008), however, it should be strongly emphasized here that our
trends are estimated without any preceding time-series homogenization and the formal errors of the
trend estimates are underestimated by a factor 2-4 (Nilsson and Elgered, 2008).
Table 6Table 6 summarizes the statistics of estimated trend differences at all 172 stations, always
between particular variants as defined in Section 5.5. Interestingly, the most significant impact is
observed due to the non-tidal atmospheric loading effects reaching differences up tobelow ±0.55
mm/year in ZTD trends for some extreme cases from the ensemble of 172 stations, and an overall 1-

sigma scatter of 0.50 mm/year from the ensemble of stations. Changes in elevation cut-off angle, particularly from 3° to 10°, reveal also a significant impact characterized by differences ~~up to~~below ±0.34 mm/year and the scatter of 0.32 mm/year. The impact of mapping function on trend estimates remains small, with a maximum difference of 0.12 mm/year and the 1-sigma scatter below 0.08 mm/year, while other strategy changes, due to time resolution of tropospheric gradients and higher-order ionospheric effects, remains negligible, always below ±0.04 mm/year for all 172 stations, with the scatter of the same magnitude. All mean biases over differences stay also below 0.05 mm/year. These results are consistent with a study performed by Ning and Elgered (2012) spaning a broader span of cutoff angles. They demonstrated a significant impact of this parameter on Integrated Water Vapor trend estimates.

Finally, we selected 12 stations ~~optimally~~ available over the entire 2nd re-processing period ~~and a~~. All estimated trends are displayed in Figure 16~~Figure 16~~. ~~Trends for 12 stations range~~ ranging from -0.05 to 0.38 mm/year. ~~with formal errors of 0.02-0.04 mm/year. It should be noted the formal errors are underestimated by a factor of 2-4 because the noise is assumed white (Nilsson and Elgered (2008).~~ Consistent with the overall results reported in Table 5~~For the 12 selected stations~~, the most significant impact for the selected 12 stations is observed in the change of elevation cut-off angle (GO2, GO3 vs. GO1) and atmospheric loading (GO4 vs. GO1) when reaching differences up to 0.1 mm/year in estimated ZTD trends. ~~A similar, but more extensive study, was performed by Ning and Elgered (2012) for Integrated Water Vapor content (IWV), roughly equal to 1/6 (ZTD-ZHD) kg.m⁻², using larger differences in the elevation cut-off angle and obtaining highly sensitive results in term of estimated IWV trends.~~ Impacts of other strategies are generally below 0.05 mm/year – variants GO4, GO5, and GO6 are very similar, but not consistent again with GO1, meaning the non-tidal atmospheric loading has a significant impact on trend estimates for selected stations with the longest data time-series.

# 7  Conclusions

In this paper, we present results of the new GOP reanalysis of all stations within the EUREF Permanent network during the period of 1996-2014. This reanalysis was completed during the 2nd EUREF reprocessing to support the realization of a new European terrestrial reference system. In the 2nd reprocessing, we focused on analysing a new product – GNSS tropospheric parameter time-series for applications to climate research. To achieve this goal, we improved our strategy for combining tropospheric parameters at midnights and at transitions in GPS weeks. We also performed seven solution variants to study optimal troposphere modelling; we assessed each of these variants in terms of their coordinate repeatability by using internal evaluations of the applied models and strategies. We also compared tropospheric ZTD and tropospheric horizontal gradients with independent evaluations obtained by numerical weather reanalysis via the ERA-Interim.

Results of the GOP Repro2 yielded improvements of approximately 50% and 25% for their horizontal and vertical component repeatability, respectively, when compared to those of the GOP Repro1 solution. Vertical repeatability was reduced from 4.14 mm to 3.73 mm when using the VMF1 mapping function, a priori ZHD, and non-tidal atmospheric loading corrections from actual weather data. Increasing the elevation cut-off angle from 3° to 7°/10° increased RMS errors of residuals from these coordinates' repeatability. All of these factors were also confirmed by the independent assessment of tropospheric parameters using NWM reanalysis data.

We particularly recommend using low-elevation observations along with the VMF1 mapping function,
as well as using precise a priori ZHD values together with the consistent model of non-tidal atmospheric
loading. While estimating tropospheric horizontal linear gradients improves coordinates' repeatability,
6-hour sampling without any absolute or relative constraints revealed a loss of stability due to their
correlations with other parameters. On the other hand, 24-h piecewise linear gradients did not indicate
a worse repeatability of coordinates estimates. For saving the time needed for the processing of 4
times less gradient parameters, we could recommend as sufficient using unconstrained 24-h piecewise
model for the first-order tropospheric asymmetry.
The impact of processing variants on long-term ZTD trend estimates was assessed at 172 EUREF
stations with time-series longer than 10 years. The most significant impact was observed due to the
non-tidal atmospheric loading effect reaching differences ~~up to~~below ±0.55 mm/year in ZTD trends for
some extreme cases from the ensemble of 172 stations. Changes in elevation cut-off angle, particularly
from 3° to 10°, revealed also a significant impact reaching differences ~~up to~~below ±0.35 mm/year. The
change of mapping function was observed rather small, with a maximum difference of 0.12 mm/year,
while other strategy changes, due to time resolution of tropospheric gradients and higher-order
ionospheric effects, remained negligible, always below ±0.04 mm/year for all 172 stations.
Assessing the tropospheric horizontal gradients with respect to the ERA-Interim reanalysis data
revealed some long-term systematic behaviour linked to degradation in antenna tracking quality. We
presented an extreme case at the Mallorca station (MALL), in which gradients systematically increased
up to 5 mm from 2003-2008 while pointing in the direction of prevailing observations at low elevation
angles. However, these biases disappeared when the malfunctioning antenna was replaced. More
cases similar to this, although less extreme, have indicated that estimated tropospheric gradients are
extremely sensitive to the quality of GNSS antenna tracking, thus suggesting that these gradients can
be used to identify problems with GNSS data tracking in historical archives.
One of the main difficulties faced during the 2[nd] reprocessing was that of the quality of the historical
data, which contains a large variety of problems. We removed data that caused significant problems
in network processing when these could not be pre-eliminated from normal equations during the
combination process without still affecting daily solutions. To provide high-accuracy, high-resolution
GNSS tropospheric products, the elimination of such problematic data or stations is even more critical
considering the targeting static coordinates on a daily or weekly basis for the maintenance of the
reference frame or the derivation of a velocity field. Before undertaking the 3[rd] EUREF reprocessing,
which is expected to begin after significant improvements have been made to state-of-the-art models,
products and software, we need to improve data quality control and clean the EUREF historical archive
in order to optimize any future reprocessing efforts and to increase the quality of tropospheric
products. These efforts should also include the collection and documentation of all available
information from each step of the 2[nd] EUREF reprocessing, including individual contributions, EUREF
combinations, time-series analyses and coordinates, and independent evaluations of tropospheric
parameters.

## Acknowledgments

The reprocessing effort and its evaluations were supported by the Ministry of Education, Youth and Science, the Czech Republic (projects LD14102 and LO1506). We thanks two anonymous reviewers and Dr. Olivier Bock for comments and suggestions which helped us to improve the manuscript.

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

**Table 1: Characteristics of GOP reprocessing models**

| Processing options | Description |
|---|---|
| Products | CODE precise orbit and earth rotation parameters from the 2nd reprocessing. |
| Observations | Dual-frequency code and phase GPS observations from L1 and L2 carriers. Elevation cut-off angle 3°, elevation-dependent weighting $1/\cos^2$ (zenith), double-difference observations and with 3-minute sampling rate. |
| Reference frame | IGb08 realization, core stations set as fiducial after a consistency checking. Coordinates estimated using a minimum constraint. |
| Antenna model | GOP: IGS08_1832 model (receiver and satellite phase centre offsets and variations). |
| Troposphere | A priori zenith hydrostatic delay/mapping function: GPT/GMFh (GO0) and VMF1/VMF1h (GO1-GO6). Estimated ZWD corrections every hour using VMF1 wet mapping function; 5 m and 1 m for absolute and relative constraints, respectively. Estimated horizontal NS and EW tropospheric gradients every 6 hours (GO0-GO5) or 24 hours (GO6) without a priori tropospheric gradients and constraints. |
| Ionosphere | Eliminated using ionosphere-free linear combination (GO0-GO6). Applying higher-order effects estimated using CODE global ionosphere product (GO5). |
| Loading effects | Atmospheric tidal loading and hydrology loading not applied. Ocean tidal loading FES2004 used. Non-tidal atmospheric loading introduced in advanced variants from the model from TU-Vienna (GO4-GO6). |


**Table 2: GOP solution variants for the assessment of selected models and settings**

| Solution ID | Specific settings and differences | Remarks and rationales |
| --- | --- | --- |
| GO0 | GMF and 3° cut-off | Legacy solution for Repro1 |
| GO1 | VMF1 and 3° cut-off | New candidate for Repro2 |
| GO2 | =GO1; 7° cut-off | Impact of elevation cut-off angle |
| GO3 | =GO1; 10° cut-off | Impact of elevation cut-off angle |
| GO4 | =GO1; atmospheric loading | Non-tidal atmospheric loading applied |
| GO5 | =GO4; higher-order ionosphere | Higher-order ionosphere effect not applied |
| GO6 | =GO4; 24-hour gradients | Stacking tropospheric gradients to 24-hour sampling |


734        **Table 3: Comparison of GOP solution variants for north, east and up coordinate repeatability.**

| Solution | North RMS [mm] | East RMS [mm] | Up RMS [mm] |
|---|---|---|---|
| GOP-Repro1/IGS05 | 3.01 | 2.40 | 5.08 |
| GOP-Repro1/IGS08 | 2.64 | 2.21 | 4.94 |
| GO0 | 1.20 | 1.30 | 4.14 |
| GO1 | 1.23 | 1.33 | 3.97 |
| GO2 | 1.24 | 1.33 | 4.01 |
| GO3 | 1.26 | 1.34 | 4.07 |
| GO4 | 1.14 | 1.24 | 3.73 |
| GO5 | 1.14 | 1.24 | 3.73 |
| GO6 | 1.14 | 1.24 | 3.73 |


**Table 4: Statistics (bias and standard deviations) of ZTD and tropospheric gradients from the seven reprocessing variants**
**compared to those obtained from the ERA-Interim NWM reanalysis. In addition to the statistics, 1-sigma range over**
**ensemble of stations is provided.**

| Solution | ZTD bias [mm] | ZTD sdev [mm] | EGRD bias [mm] | EGRD sdev [mm] | NGRD bias [mm] | NGRD sdev [mm] |
|---|---|---|---|---|---|---|
| GO0 | -1.5 ± 2.1 | 8.8 ± 2.0 | -0.04 ± 0.08 | 0.39 ± 0.10 | +0.01 ± 0.09 | 0.43 ± 0.12 |
| GO1 | -2.0 ± 2.1 | 8.3 ± 2.2 | -0.04 ± 0.08 | 0.39 ± 0.10 | +0.01 ± 0.09 | 0.42 ± 0.13 |
| GO2 | -1.9 ± 2.2 | 8.4 ± 2.2 | -0.05 ± 0.10 | 0.41 ± 0.10 | +0.00 ± 0.12 | 0.45 ± 0.12 |
| GO3 | -1.8 ± 2.3 | 8.5 ± 2.1 | -0.08 ± 0.13 | 0.43 ± 0.11 | -0.01 ± 0.14 | 0.49 ± 0.12 |
| GO4 | -1.8 ± 2.4 | 8.1 ± 2.1 | -0.04 ± 0.09 | 0.38 ± 0.10 | +0.00 ± 0.09 | 0.40 ± 0.12 |
| GO5 | -1.8 ± 2.4 | 8.1 ± 2.1 | -0.05 ± 0.09 | 0.38 ± 0.10 | +0.01 ± 0.08 | 0.40 ± 0.12 |
| GO6 | -1.8 ± 2.4 | 8.2 ± 2.1 | -0.04 ± 0.08 | 0.29 ± 0.06 | +0.01 ± 0.09 | 0.28 ± 0.06 |


**Table 5: Median, minimum (min) and maximum (max) values of total ZTD biases and standard deviation (sdev) over all**
**stations. Units are millimetres.**

| Compared variants | ZTD bias median | ZTD bias min | ZTD bias max | ZTD sdev median | ZTD sdev min | ZTD sdev max |
|---|---|---|---|---|---|---|
| GO1-GO0 | -0.36 | -1.52 | +0.70 | 2.01 | 0.69 | 3.82 |
| GO2-GO1 | +0.03 | -0.81 | +1.66 | 0.66 | 0.15 | 1.29 |
| GO3-GO1 | +0.03 | -2.22 | +2.66 | 1.10 | 0.31 | 2.04 |
| GO4-GO1 | +0.05 | -3.29 | +5.55 | 1.37 | 0.68 | 4.72 |
| GO5-GO4 | -0.02 | -0.31 | +0.07 | 0.07 | 0.04 | 0.30 |
| GO6-GO4 | -0.02 | -0.23 | +0.16 | 1.24 | 0.76 | 2.46 |


**Table 6: Mean statistics of ZTD trends differences estimated between variants for 172 stations. Units are millimetres/year.**

| Statistics | GO1-GO0 | GO2-GO1 | GO3-GO1 | GO4-GO1 | GO5-GO4 | GO6-GO4 |
|---|---|---|---|---|---|---|
| Min | -0.118 | -0.141 | -0.308 | -0.547 | -0.017 | -0.038 |
| Max | 0.045 | 0.179 | 0.331 | 0.452 | 0.031 | 0.036 |
| mean | 0.036 | 0.018 | 0.012 | -0.048 | 0.007 | 0.001 |
| Sdev | 0.081 | 0.160 | 0.319 | 0.499 | 0.024 | 0.037 |




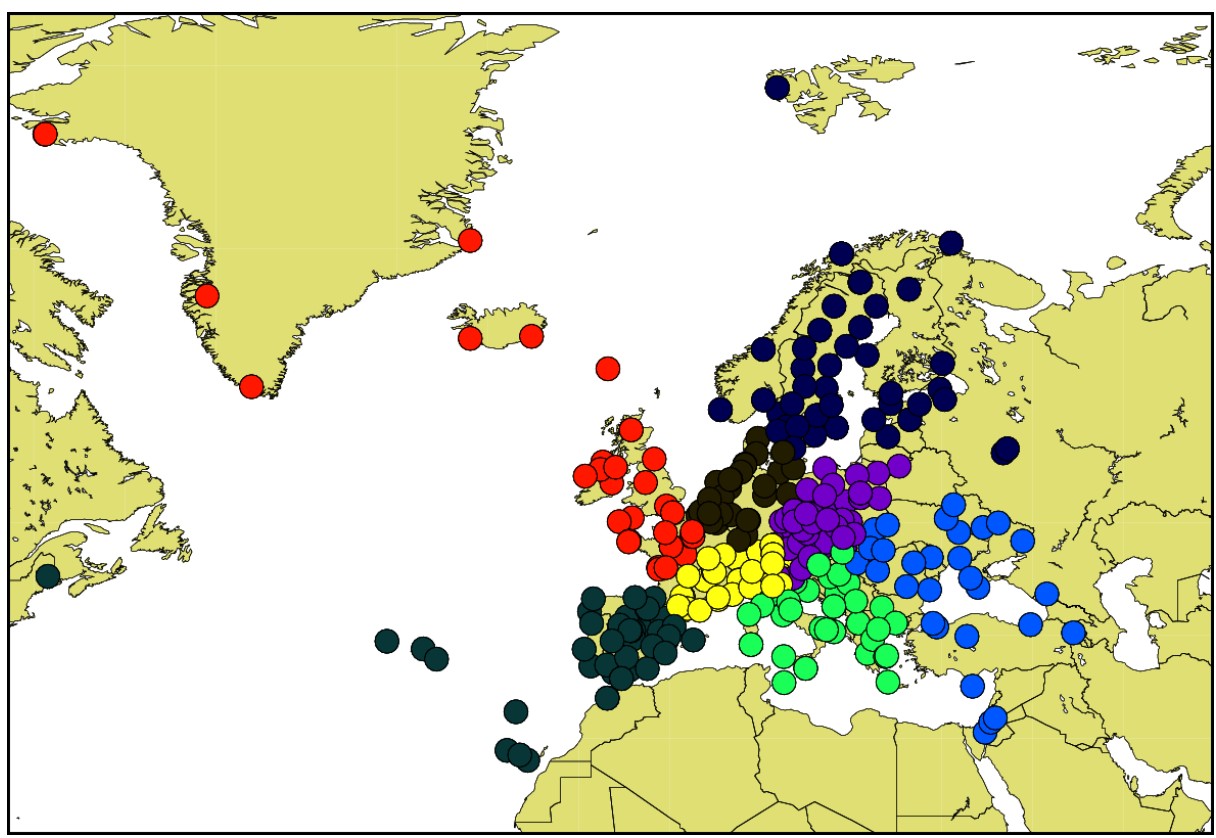

**Figure 1: EUREF Permanent Network's clusters (designated by different colours) in the 2ⁿᵈ GOP reprocessing.**


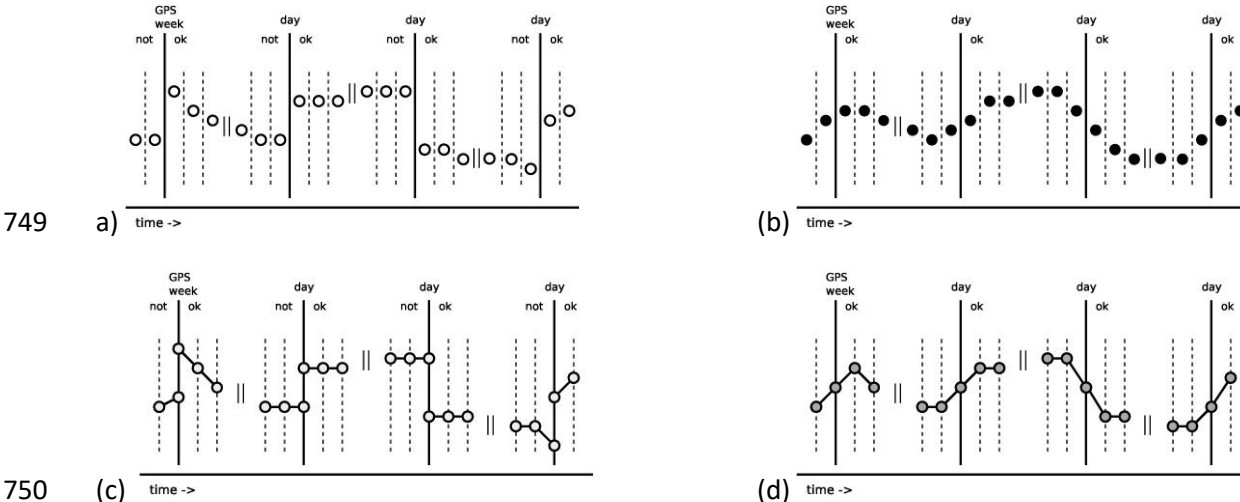

a)
(c)    (d)
**Figure 2: Charts of 4 variations on representations of tropospheric parameters. Right (b), (d) and left (a), (c) panels**
**display estimates made with and without midnight combinations, respectively. Top (a), (b) and bottom (c), (d) panels**
**display the piecewise constant and the linear model, respectively.**

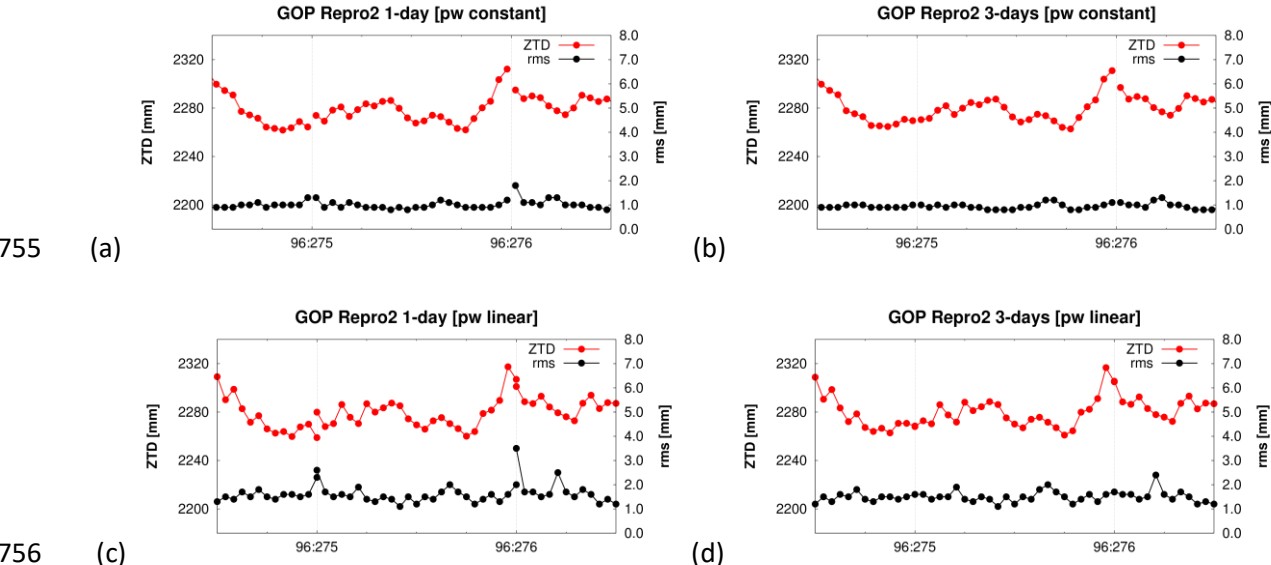

(a)    (b)
(c)    (d)
Figure 3: Four variations in representation of tropospheric parameters. Right (b), (d) and left (a), (c)
panels display estimates with and without midnight combinations, respectively. Top (a), (b) and
bottom (c), (d) panels display the piecewise constant and the piecewise linear model, respectively.

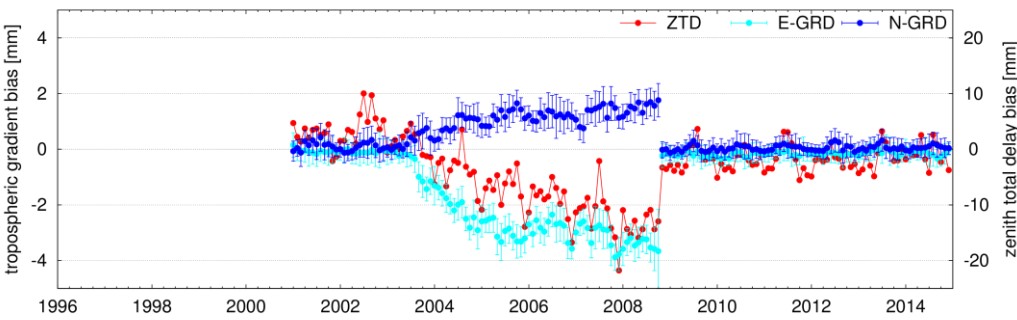

**Figure 4: MALL station - monthly mean differences in tropospheric horizontal gradients with respect to the ERA-Interim.**


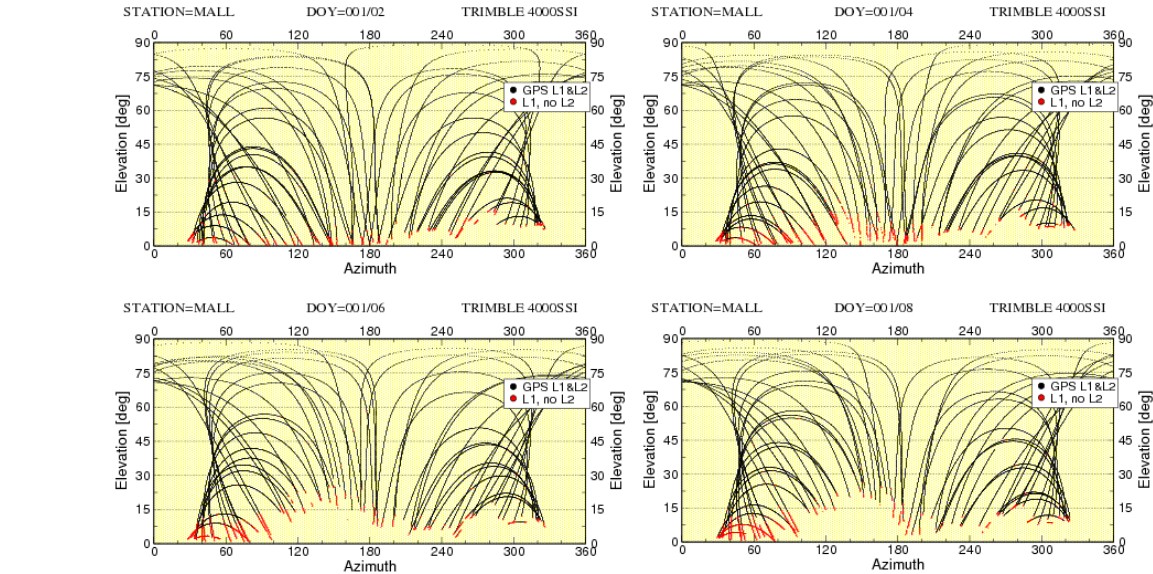

**Figure 5: Low-elevation tracking problems at the MALL station during the period of 2003-2008. From left-top to right-**
**bottom: January 2002, 2004, 2006 and 2008 (courtesy of the EPN Central Bureau, ROB).**

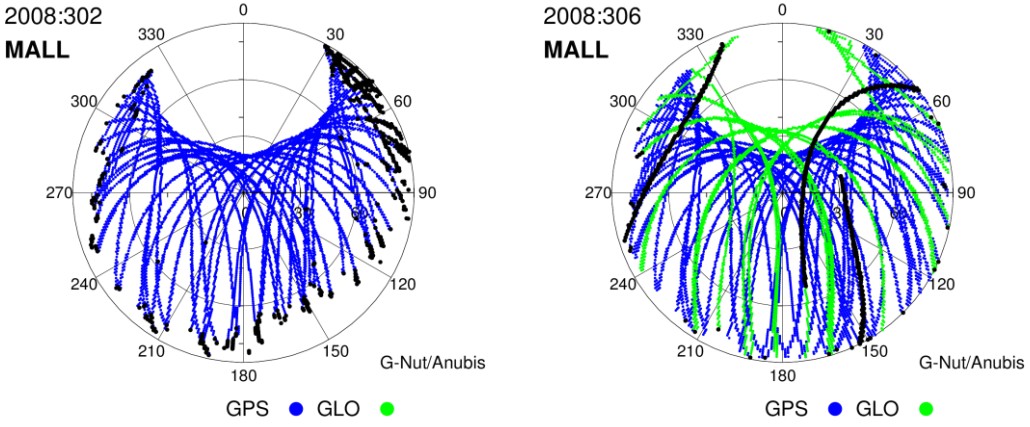


**Figure 6: Sky plots before (left) and after (right) replacing the malfunctioning antenna at the MALL site (Oct 30, 2008). Black dots indicates single-frequency observations available only.**

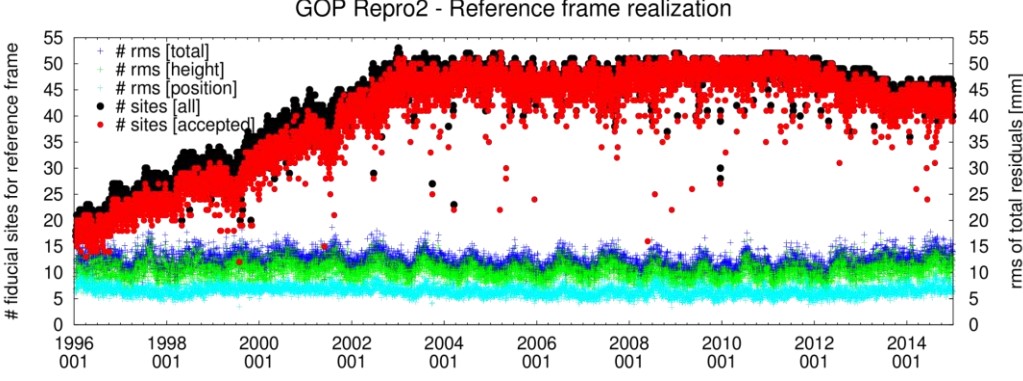


**Figure 7: Statistics of the daily reference system realization: a) RMS of residuals at fiducial stations (representing the**
**total, height and position); b) number of stations (all and accepted after an iterative control)**

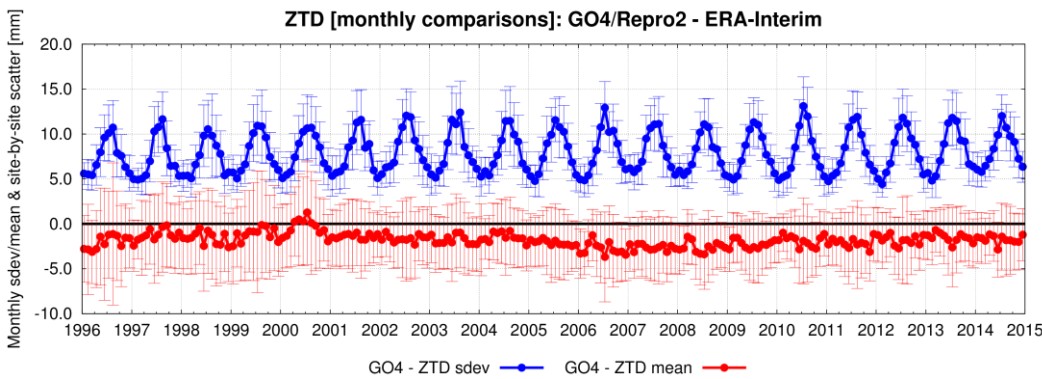


**Figure 8: Monthly means of bias and standard deviation of official GOP ZTD product compared to those of the ERA-Interim.**
**Error bars indicate standard errors of mean values over all compared stations.**


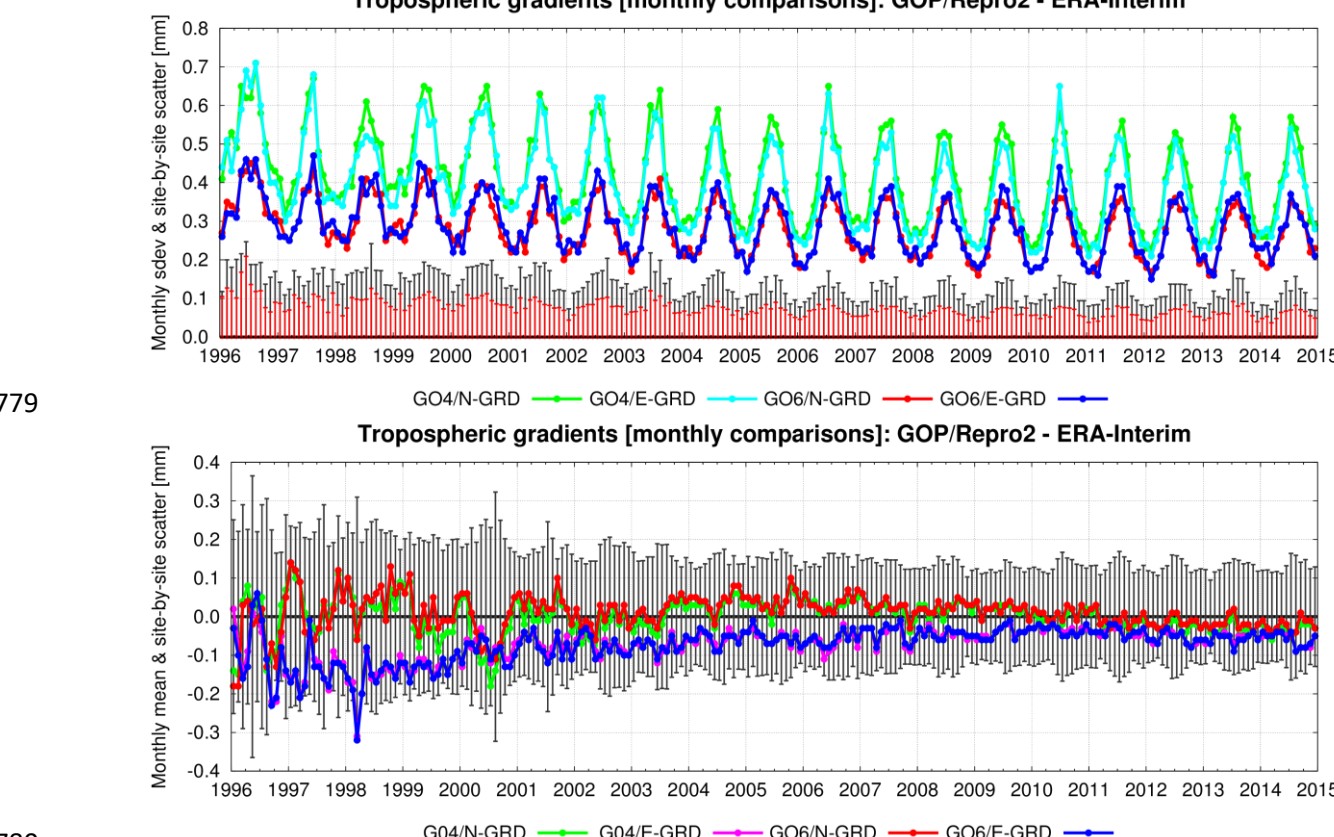


**Figure 9: Monthly means of bias and standard deviation of tropospheric horizontal north (N-GRD) and east (E-GRD)**
**gradients compared to those obtained by ERA-Interim. Note: Similar products are almost superposed. Error bars indicate**
**standard errors of mean values over all compared stations plotted from the zero y-axis to emphasise seasonal variations**
**and trends. Error bars are displayed for north gradients only, however, being representative for the east gradients too.**

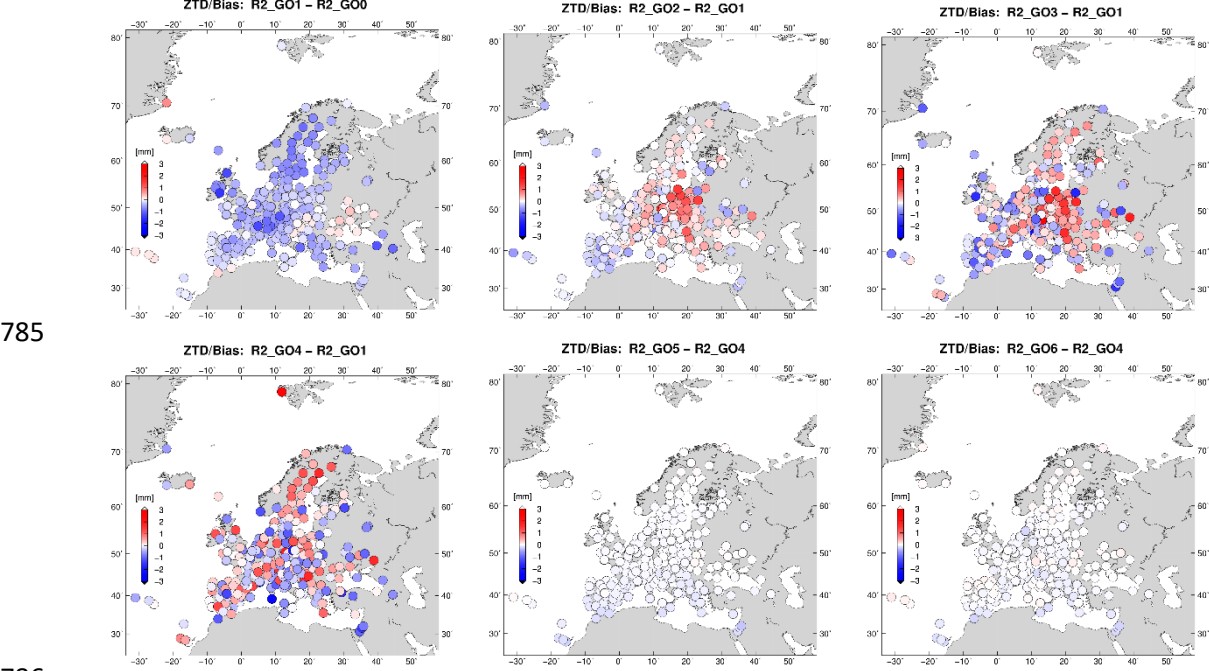

**Figure 10: Geographic visualization of biases from inter-comparisons of GOP 2$^{nd}$ reprocessing variants.**

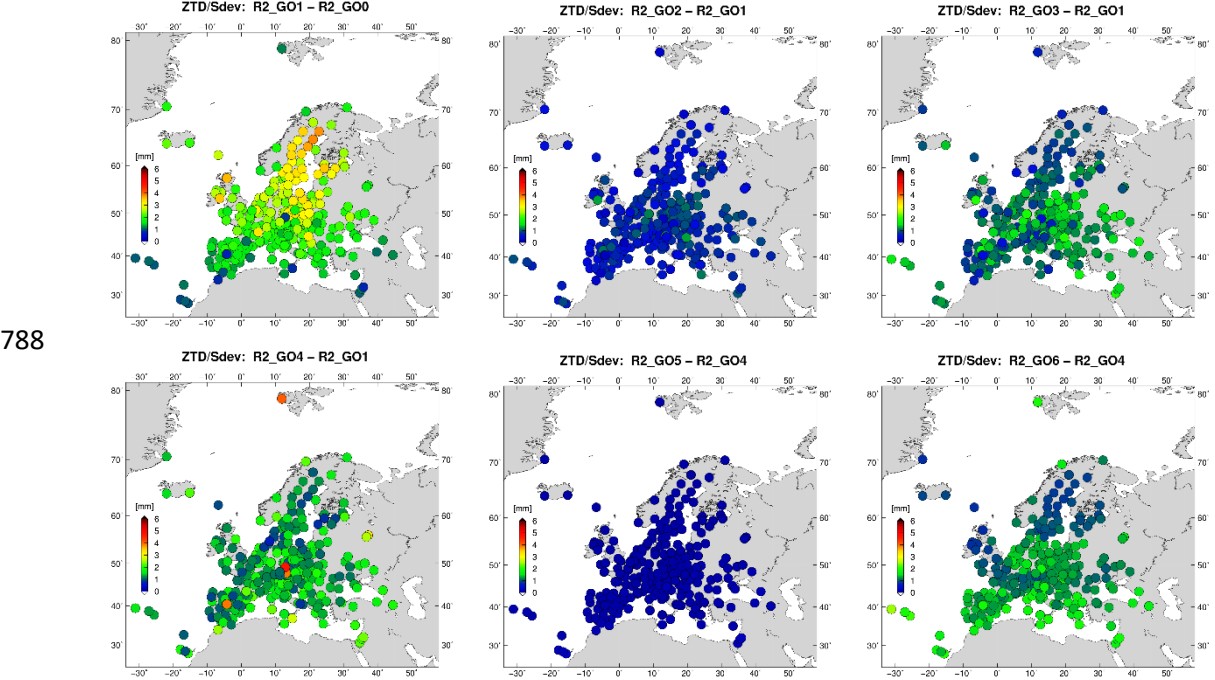


**Figure 11: Geographic visualization of standard deviations from inter-comparisons of GOP 2nd reprocessing variants.**

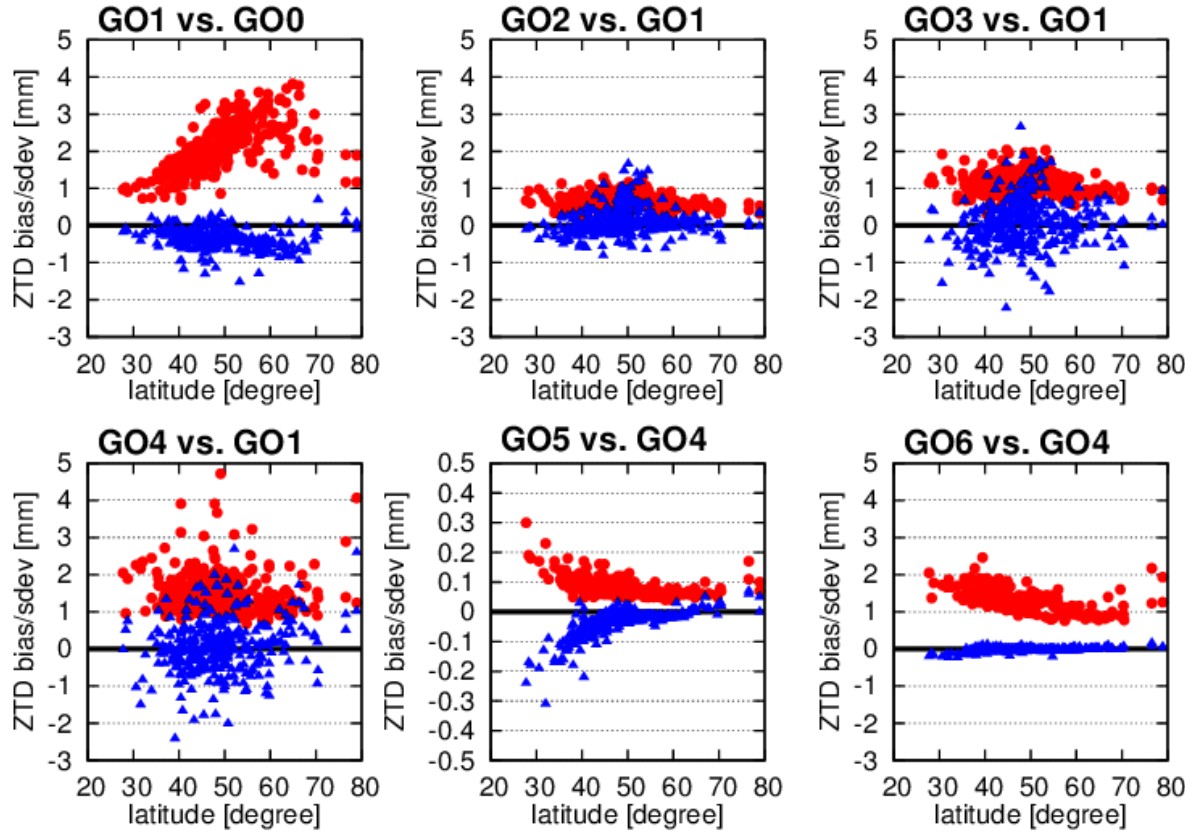


**Figure 12: Dependence of ZTD biases (blue) and standard deviations (red) from inter-comparisons of GOP 2nd**
**reprocessing solution variants on station latitude. Note different y-range for the GO5 vs. GO4 comparison.**

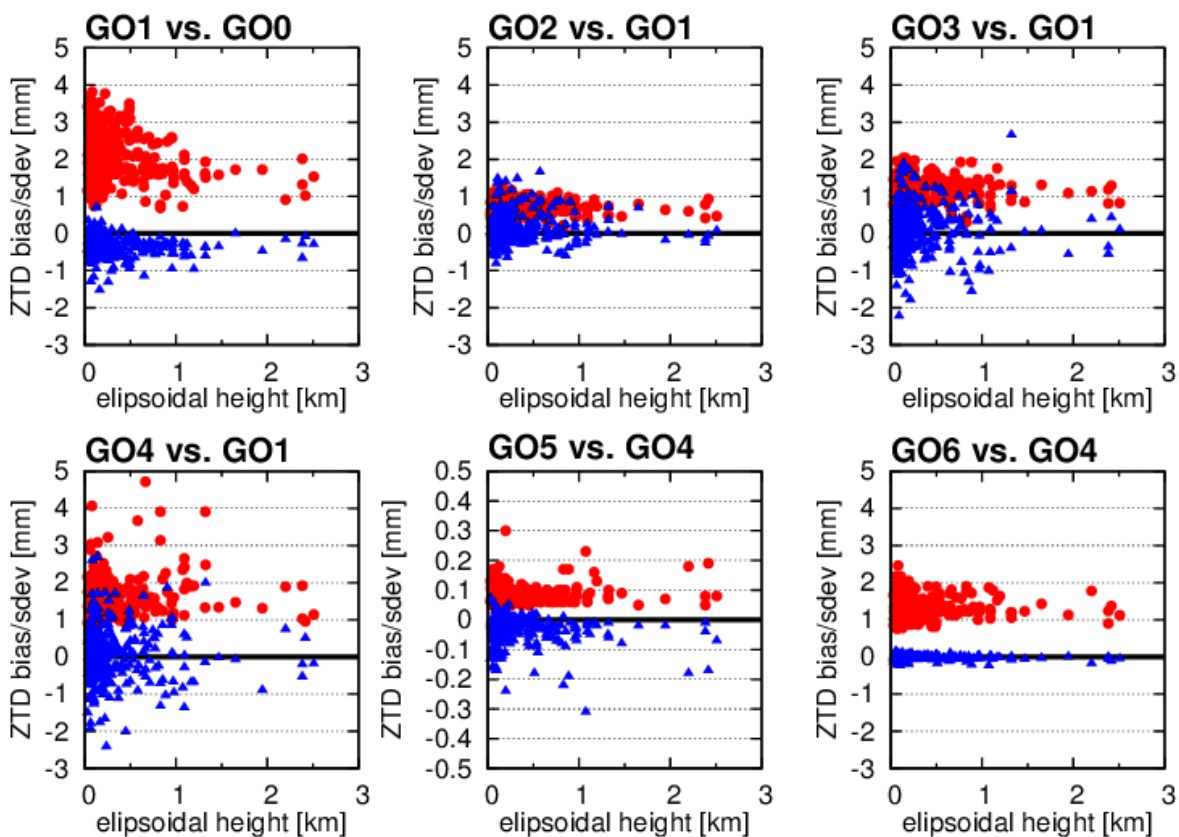


**Figure 13: Dependence of ZTD biases (blue) and standard deviations (red) from inter-comparisons of GOP 2$^{nd}$ reprocessing**
**solution variants on station ellipsoidal height. Note different y-range for the GO5 vs. GO4 comparison.**

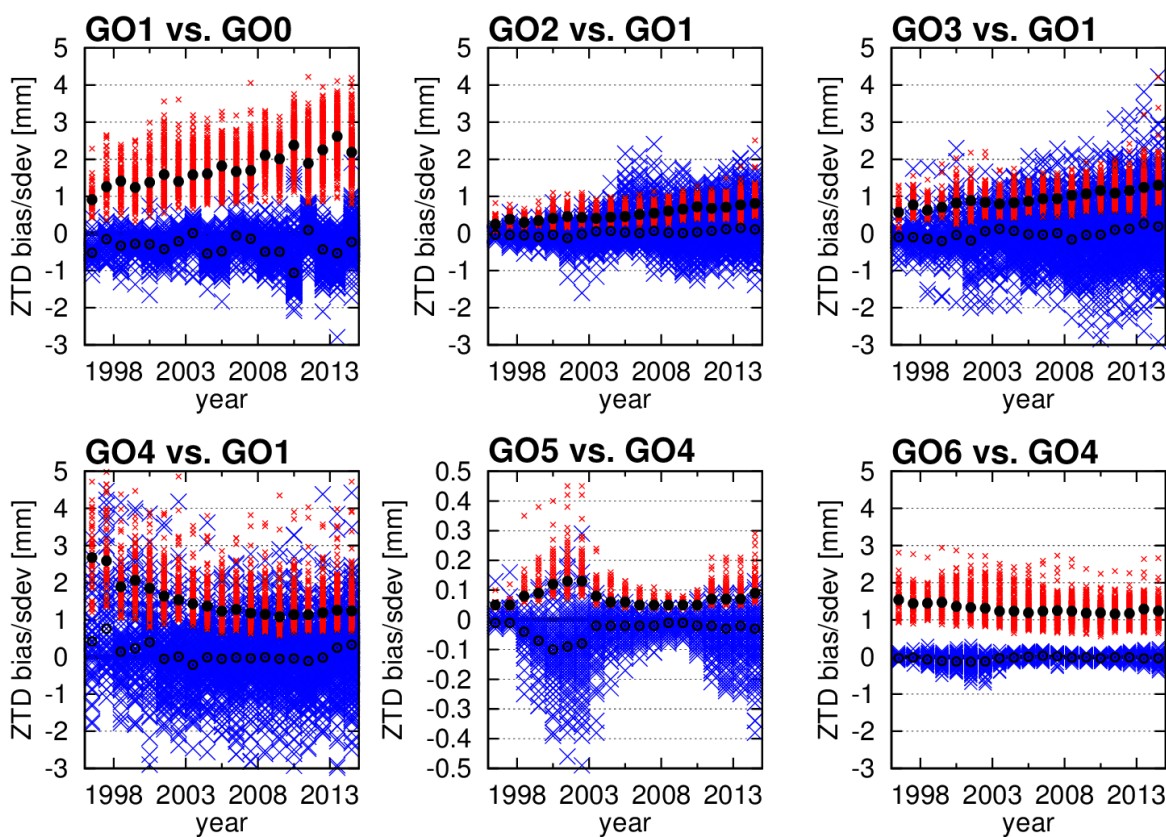

Figure 14: Dependence of ZTD biases (blue), mean biases (unfilled black circles), standard deviations (red) and mean standard deviations (filled black circles) from inter-comparisons of GOP 2nd reprocessing solution variants on year. Note different y-range for the GO5 vs. GO4 comparison.

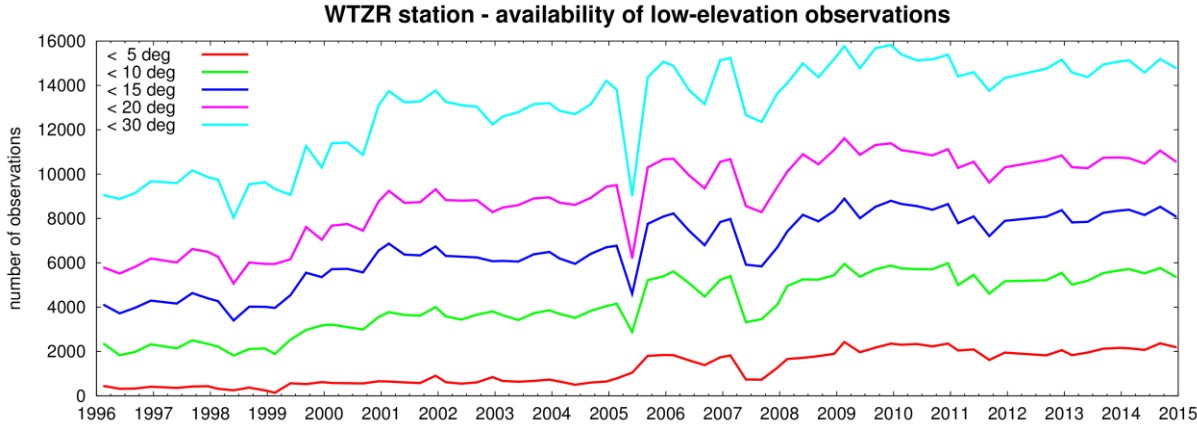


**Figure 15: Availability of observations at low-elevation angles (below 5°, 10°, 15°, 20° and 30°) for WTZR station.**

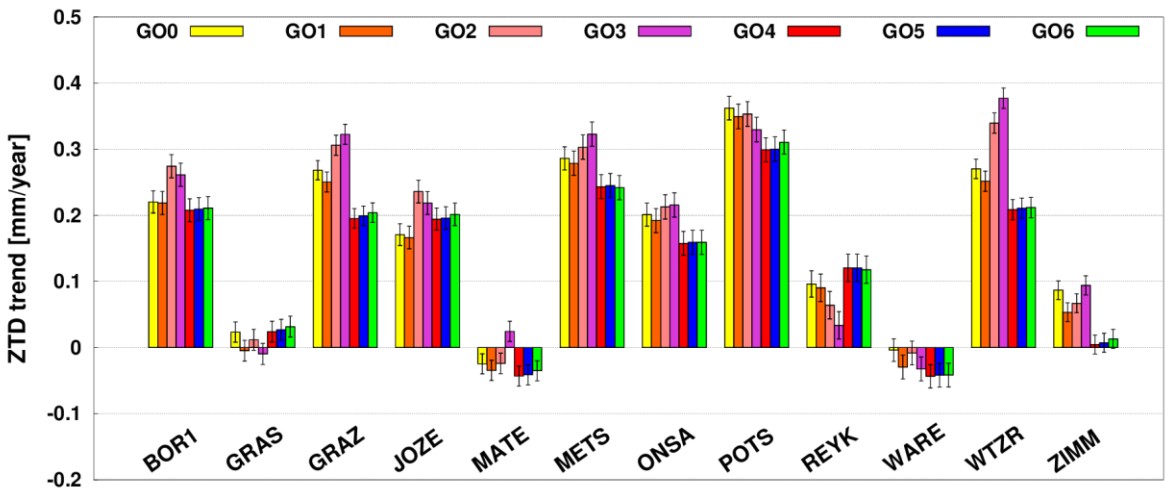


**Figure 16: Long-term ZTD trend estimates and their formal errors (error bars) for all processing variants**