# Peer review of "Tropospheric products of the 2nd GOP European GNSS reprocessing (1996-2014)"

_Atmospheric Measurement Techniques, 2017_

## Referee Comment (RC1) · Anonymous Referee #1 · 8 Mar 2017

Dousa and Vaclavovic provide a clear and systematic report on the tropospheric products from a reprocessing of the EUREF network. They use both internal metric - the RMS of coordinate repeatabilities - as well as an external comparison with numerical weather model reanalyses to assess the performance of the standard reprocessing approach as well as a suite of variants designed to test the impact of possible alternative strategies. They are able to validate a significant improvement in the reprocessed time-series and identify recommendations for optimal performance in the future processing. The most notable result from their work is that a low - 3-degree - cutoff angle for the data provides the best results. Although this is expected to improve the geometric distribution and help decorrelate the vertical position and troposphere parameters, the mapping functions and antenna models are not expected to be well described for very low elevation angles. It is therefore noteworthy that this processing strategy improved

the results.

Although their results might not be ground-breaking, they are carefully performed and provide valuable information to a range of researchers. With a few minor modifications I recommend this for acceptance and publication.

There are a few areas that I think should be cleaned up or further explained in order to ensure that (particularly non-specialists) can better understand the work and its implications: 1) Section 2: I assume, though it is not stated, that a set of (global?) IGS sites were included in the processing of each of the sub-networks, both to ensure strong ties to IGS08 as well as provide the long-baselines necessary to ensure retrieval of absolute ZTDs?

2) Section 3: It is definitely important that artifacts at day-boundaries are mitigated, and the strategy followed to ensure ZTD continuity seems perfectly good. The smoothness of the RMS across midnight is a good indicator that the strategy is performing as expected, it would be nice, however, to see a quantitative validation of the positive impact. Can this be pulled out of the ERA-Interim comparisons?

3) Figure 4: I am uncertain what the panel titles mean? In the text (line 173) this figure is referenced as Figure 3, and suggests the data are from 1999, though the x-axis labels indicate 1996?

4) Seasonality of coordinate RMS (Figure 5 & lines 228-230). Some statement about the source of the seasonality seems called for. This may include a citation where this has been previously described, though it might also/instead reference Figure 7 which indicated that the tropo gradients show a similar seasonality suggesting that the limiting factor may be in the modeling of the atmosphere (rather than a seasonal source(s) of increased ground motion).

Minor comments: Section 4.2: discussion first of Table 5 and then Figures 6 & 7 (lines 270-324) means jumping back and forth between ZTD and tropo gradients. I would

suggest re-ordering the discussion to address the ZTD portion of Table 5 along with Figure 6 and then the gradient section and Figure 7.

Gradient units: tropo gradients are not entirely straightforward to describe - and it is not clear in the paper what the numbers given actually represent, and strictly speaking the units of the gradients in ZTD (or ZWD) cannot be mm... so how are the tropo gradients being implemented, and what do the values given physically mean?

Suggestions: Abstract, line 24. "assessing" = "comparing"?

Intro, line 33. "... (GPS) became fully operational in 1995..."

Intro, line 85. "enhance" = "improve"?

Section 3. Line 155. "... an interval, or b) by..."

Section 3. Line 165. "Finally, we represented the piece-wise linear solutions in terms of offsets,..."

Section 4.2 lines 241-242. Not quite sure what/where the "yield values" are refering to? This sentence needs a little work for clarity. Section 4.2 line 249: "...as the same blind mapping function and a priori ZHD values are used for both the GO0 and..."

---

## Referee Comment (RC2) · Anonymous Referee #2 · 17 Mar 2017

The manuscript presents results of the 2nd reprocessing of the EPN network performed by GOP analysis centre. Seven variants of processing were carried out and compared to GOP-Repro1 and combined EUREF Repro 1 solutions. Moreover, independent data from the ERA-Interim global reanalysis were used to validate tropospheric products from GNSS processing. Authors assessed all solutions in term of repeatability of station coordinates and also analysed biases and standard deviations of the derived ZTDs and horizontal gradients. They also discussed the relationship between tropospheric gradient bias and antenna tracking. In my opinion this paper is well written and will be of interest of AMT readers. However, I have some concerns and questions before it can be accepted for publication.

1. Did you do any screening of coordinates and ZTD/gradients obtained from your processing? If so, what was the screening procedure?

2. In this paper almost all analysis and statistics (expect interesting case with MALL station) are quite general. You may want to try to analyse the results in more details and try to find cases when change of the processing parameters had the clear influence on the estimated coordinates and tropospheric parameters. E.g. you could do more careful analysis and consider possible dependence on the on the localization of stations, antenna models, etc. (for example maybe for stations in high mountains or closer to equator some variant are better than others). This would improve the content of the manuscript.

3. Section 2, line 106, Figure 1:

a) You wrote that the network was split into 10 sub-networks. In Figure 1 based on different colours I can distinguish only 6 clusters. It is better to change the markers and e.g. some clusters mark as squares.

b) Did you use common stations to link the clusters in the network solution?

c) How did you choose the clusters of stations? Based on the localization of the stations? I can see in Figure 1 that clusters are regional – stations which are located close to each other are in the same cluster, and the stations of each sub-networks are always the same. Is it an optimal solution of the sub-networks design? Santamaría-Gómez (2010) showed the results of processing of global network clustered into "dynamic subnetwork", where closer stations were distributed in different sub-networks in order to obtain a regular distribution based on station baseline geometry. They showed a noticeable improvement in the percentage of fixed ambiguities, especially before the year 2000, and also improvement of position repeatability and transformation parameters with respect to a "static sub-networks" solution. Did you test maybe this kind of clustering in your processing?

- Santamaría-Gómez, A. (2010), Estimation of crustal vertical movements with GPS in a geocentric frame, within the framework of the TIGA project, doctoral dissertation of the Observatoire de Paris.

AMTD
- Santamaría-Gómez, A.; Bouin, M.-N.; Wöppelmann, G. (2009), Impact of subnetwork configuration on global scale GPS processing, EGU General Assembly 2009.

4. Section 4.1, lines 202-223: You wrote that you used an interactive procedure of validation of the fiducial stations. Can you be more specific on what this procedure was and how it works? Did you choose stations based on daily repeatability of their coordinates? What was your set of fiducial stations? IGS stations?

5. Section 4.2:

a) lines 233-253: It's a quite long paragraph about comparison of ZTD obtained from GOP Repro2 reprocessing to EUREF Repro1 products. We can expect that EUREF Repro 1 is worse than each version of Repro 2. The fact that some variant of the reprocessing is closer to EUREF Repro 1 does not mean that it is better. So, is it really useful to show such results? Does it bring any meaningful statistics? I think comparison to any external data (for instance ERA-Interim what is shown in next paragraph) is more interesting and conclusive.

b) lines 260-274, Figure 5: GNSS ZTD from each reprocessing compared to ZTD from ERA-Interim is characterized by a negative bias. We can also notice it in the EPN solution. Can you explain why the bias is negative?

6. References, line 472: Please, change reference Pacione et al. (2017) to: Pacione, R., Araszkiewicz, A., Brockmann, E., and Dousa, J.: EPN Repro2: A reference GNSS tropospheric dataset over Europe, Atmos. Meas. Tech. Discuss., doi:10.5194/amt-2016-369, in review, 2016.

---

## Editor Comment (EC1) · O. BOCK (Editor) · 17 Apr 2017

Dear authors, please find below a few additional comments to those provided by the two referees.

1. You write that the EUREF recommendations are followed which specify that "weekly coordinates should be used to estimate tropospheric parameters on a daily basis" (L129) and that the coordinates were fixed to these values (L144). Did you fix the coordinates for all stations or only the fiducial stations? Fixing station heights is known to produce biases in ZTD estimates due to un-modelled station motions (tidal and non-tidal, e.g. seasonal) and other error sources (because of the correlation between estimated parameters). Abrupt changes and drifts over time that impact stations height would then also map into ZTD estimates. Can you comment on the uncertainty in the

[Figure]

ZTD estimates, and possibly also gradients, due your specific processing strategy?

2. The relevance of this study is that several processing variants are produced with the same software. The results are thus not obscured by inter-software biases. However, the discussion of results from the different variants is quite short in the manuscript. The accuracy of tropospheric parameters is only analysed based on Table 5 and Figure 5 and 6. Table 5 compares the biases and standard deviations over all stations and all times for the different variants. It is striking that the differences in these numbers are tiny. I would not be surprised that a spatialized analysis reveals significant impact of changing the cutoff angle and mapping functions at sites in different climatic regions, in mountainous areas, or close to the sea. If relevant, I suggest that you complement the paper with spatialized results.

3. Table 4 comparing the GOP solutions to the outdated EUREF repro1 is not relevant. This comparison might be done as an initial consistency check of the new solutions compared to the legacy EUREF reference. I thus suggest removing this figure and the related text from the manuscript.

4. The temporal homogeneity of long time series is crucial when trends are to be estimated. Given that there is presently a high interest of the GNSS/climate community in estimating trends, I think it would be useful to complement the results with an analysis of trends for the different processing variants. There are many questions like: which cutoff angle and mapping functions choose to get the most homogenous time series? What is the impact of changing quality in GNSS observations over time? Again, the conclusions might be station dependent and both overall and spatialized analyses might be necessary to document them properly.

5. The impact of the temporal resolution of gradient parameters is intriguing. Indeed, better accuracy is expected when combing the 6-hourly to 24-hourly estimates. With 4 times more observations the standard deviation is expected to be divided by a factor of 2. However, according to Table 5 the improvement is only by a factor of 1.3 suggesting there is serial correlation in the errors. Is this reduction factor is uniformly distributed over stations and stable over time? Can you be more specific about the correlation between gradients and other parameters suggested in the manuscript? (L280 and 309)

6. The drift in the gradient estimates at station MALL (Fig. 7) is impressive. How did the ZTD estimates evolve during the period when gradients drifted? Did you detect other cases like this? In the case of MALL the cause was identified as a tracking problem. Did you detect other causes which could produce such drifts in gradients or ZTDs? It would be interesting to include a check on gradients as part of a data screening method. I suggest considering this idea in the discussion.

Other specific comments are given in the annotated PDF.

Please also note the supplement to this comment:
http://www.atmos-meas-tech-discuss.net/amt-2017-11/amt-2017-11-EC1-supplement.pdf

[Figure]

**Supplement:**

[revised manuscript text omitted]

---

## Author Response (AR1)

We would like to thank for the comments which helped us to improve the manuscript significantly.

Jan Douša, Pavel Václavovic, Michal Eliaš

**Responses to Review #1**

RC1: Dousa and Vaclavovic provide a clear and systematic report on the tropospheric products from a reprocessing of the EUREF network. They use both internal metric – the RMS of coordinate repeatabilities - as well as an external comparison with numerical weather model reanalyses to assess the performance of the standard reprocessing approach as well as a suite of variants designed to test the impact of possible alternative strategies. They are able to validate a significant improvement in the reprocessed timeseries and identify recommendations for optimal performance in the future processing.

The most notable result from their work is that a low - 3-degree - cutoff angle for the data provides the best results. Although this is expected to improve the geometric distribution and help decorrelate the vertical position and troposphere parameters, the mapping functions and antenna models are not expected to be well described for very low elevation angles. It is therefore noteworthy that this processing strategy improved the results.

As it concerns the solution using the lowest elevation angle, the use of VMF1 down to 3 degrees is, in principle, not problematic from a theoretical point of view as the VMF1 has been estimated using numerical weather data ray-tracing at elevations of 3 degrees only. The mapping function is thus optimized for 3 degrees, while not always perfect for higher elevation angles. However, the impact is visible mainly between 'optimized' 3 degrees and roughly 10 degrees when mapping function is directly compared to ray-tracing at elevation angles above 3 degrees, e.g. F. Zus (2015) and D. Landskroun (2017) preparing the revised concept for VMF3 (EGU2017 presentation). A short note and the reference (F. Zus et al., 2015) were added to the manuscript (Section 4.1).

Zus F, Dick G, Dousa J, Wickert J (2015), **Systematic errors of mapping functions which are based on the VMF1 concept**, GPS solut,19(2):277-286, doi:10.1007/s10291-014-0386-4

RC1: Although their results might not be ground-breaking, they are carefully performed and provide valuable information to a range of researchers. With a few minor modifications I recommend this for acceptance and publication. There are a few areas that I think should be cleaned up or further explained in order to ensure that (particularly non-specialists) can better understand the work and its implications:

1) Section 2: I assume, though it is not stated, that a set of (global?) IGS sites were included in the processing of each of the sub-networks, both to ensure strong ties to IGS08 as well as provide the long-baselines necessary to ensure retrieval of absolute ZTDs?

The processed network consisted of EUREF Permanent Network (EPN) only, thus no global stations were included. The overall network extent is large enough (Min/Max Lat: +27°/+79°, Lon: -79°/+58°) to resolve tropospheric parameters in absolute sense and to ensure strong ties to IGS08 by many fiducial stations. The datum has been carefully revised whenever combining clusters into a full network

solution and highly reliable a priori coordinates has been carefully applied in a sub-network solutions for pre-eliminating phase ambiguities. Two short notes were added (Section 2).

2) Section 3: It is definitely important that artefacts at day-boundaries are mitigated, and the strategy followed to ensure ZTD continuity seems perfectly good. The smoothness of the RMS across midnight is a good indicator that the strategy is performing as expected, it would be nice, however, to see a quantitative validation of the positive impact. Can this be pulled out of the ERA-Interim comparisons?

Unfortunately not for GOP solutions as we haven't prepared both variants (3-day and 1-day). However, we have evaluated two global Repro2 solutions from the IGS analysis centre CODE (Centre for Orbit Determination in Europe) in GOP-TropDB previously, which included 1-day (official IGS contribution, COF) and 3-day (unofficial product, COD) solutions. The results indicated up to a several millimetres or even higher differences at midnights in ZTDs due to boundary effect. We suppose the actual effect might reach up to a centimetre or even more considering that values at the middle epoch of low-resolution ZTD products (2-hour) could have been compared only. The paragraph and reference to the CODE product and its evaluation in GOP-TropDB were added (Section 3).

3) Figure 4: I am uncertain what the panel titles mean? In the text (line 173) this figure is referenced as Figure 3, and suggests the data are from 1999, though the x-axis labels indicate 1996?

We are confused what is really questioned as line 173 has no reference to Figure 3 and we haven't found any mistake with the reference to Figure 4 in Section 4.1. However, we should clarify one possible misunderstanding: the GOP product cover the period of 1996-2014 while the year 1999 (day 209) was referred as having a minimum number (12) of fiducial stations applied for realizing of IGS08 reference frame. We slightly modified the text to be clear in this context (the last paragraph in Section 4.1).

4) Seasonality of coordinate RMS (Figure 5 & lines 228-230). Some statement about the source of the seasonality seems called for. This may include a citation where this has been previously described, though it might also/instead reference Figure 7 which indicated that the tropo gradients show a similar seasonality suggesting that the limiting factor may be in the modelling of the atmosphere (rather than a seasonal source(s) of increased ground motion).

Unfortunately, also here we have trouble to identify the correct place in the manuscript as the coordinate RMS is shown in Figure 4 and no relation was found within lines 228-230. However, we agree that a major contribution to the seasonality on coordinates (particularly height component) is due to the troposphere modelling as it is also clear from the high variability of wet zenith component (and most likely wet gradients) due to varying water vapour content. At the end of Section 4.1, we added a sentence describing the reason for seasonality in coordinate RMS and, particularly, the height component.

Minor comments: Section 4.2: discussion first of Table 5 and then Figures 6 & 7 (lines 270-324) means jumping back and forth between ZTD and tropo gradients. I would suggest re-ordering the discussion to address the ZTD portion of Table 5 along with Figure 6 and then the gradient section and Figure 7.

Thank you for suggestion, we have reorganized the text by splitting the evaluations of ZTDs and horizontal gradients into independent subsections and we believed it clarified the description. Section 4 was modified.

Gradient units: tropo gradients are not entirely straightforward to describe - and it is not clear in the paper what the numbers given actually represent, and strictly speaking the units of the gradients in

ZTD (or ZWD) cannot be mm... so how are the tropo gradients being implemented, and what do the values given physically mean?

Although are principally assumed to represent ZTD change with a distance (in North and East directions), practically, they have to be parametrized using elevation angles of individual observations instead of the distance. However, the relation between the effective distance and elevation angle requires assumptions about the effective height of the tropospheric effect. The interpretation of the tropospheric horizontal gradients in Bernese software introduces a small angle applied for the tilting the zenith direction in the mapping function with gradients representing (in unit of length) the tilting angle multiplied by the delay in zenith (Meindl et al., 2004). The description was added (Section 4.3).

Suggestions: Abstract, line 24. "assessing" = "comparing"?
corrected

Intro, line 33. "... (GPS) became fully operational in 1995..."
corrected

Intro, line 85. "enhance" = "improve"?
corrected

Section 3. Line 155. "... an interval, or b) by..."
corrected

Section 3. Line 165. "Finally, we represented the piece-wise linear solutions in terms of offsets,..."
corrected

Section 4.2 lines 241-242. Not quite sure what/where the "yield values" are referring to? This sentence needs a little work for clarity.
corrected

Section 4.2 line 249: "...as the same blind mapping function and a priori ZHD values are used for both the GO0 and..."
The sentence was finally removed from the manuscript (see other reviewer comments).

**Responses to Review #2.**

RC1: The manuscript presents results of the 2nd reprocessing of the EPN network performed by GOP analysis centre. Seven variants of processing were carried out and compared to GOP-Repro1 and combined EUREF Repro 1 solutions. Moreover, independent data from the ERA-Interim global reanalysis were used to validate tropospheric products from GNSS processing. Authors assessed all solutions in term of repeatability of station coordinates and also analysed biases and standard deviations of the derived ZTDs and horizontal gradients. They also discussed the relationship between tropospheric gradient bias and antenna tracking. In my opinion this paper is well written and will be of interest of AMT readers. However, I have some concerns and questions before it can be accepted for publication.

1. Did you do any screening of coordinates and ZTD/gradients obtained from your processing? If so, what was the screening procedure?

Within the processing, we screened station coordinate repeatabilities from weekly combined solutions and we identified any problematic station for which north/east/up residuals exceeded 15/15/30 mm or RMS of north/east/up coordinate component exceeded values 10/10/20 mm. Such station was a priori excluded from the tropospheric product for the corresponding day. There were other standard control procedures within the processing when individual station could have been excluded, e.g. if a) less than 60% of GNSS data available, b) code or phase data revealed poor quality, c) station metadata were found inconsistent with data file header information (receiver, antenna and dome names, antenna eccentricities) and, d) phase residuals were too large for all satellites in the processing period indicating a problem with station. Tropospheric parameters were estimated practically without constrains (sigma greater than 1 m) thus parameter formal errors reflect relative uncertainties of estimates. Large errors usually indicate lack of observations contributing to the parameter. During the tropoosheric parameter evaluations, we applied filter for exceeding formal errors of estimated parameters (ZTD sigma greater than 3 mm, normal cases stay below 1 mm). In monthly statistics we also applied iterative procedure for excluding residuals exceeding 3-sigma of standard deviation calculated from the compared differences (Gyori and Dousa, 2016). The description was added (end of Section 2).

2. In this paper almost all analysis and statistics (expect interesting case with MALL station) are quite general. You may want to try to analyse the results in more details and try to find cases when change of the processing parameters had the clear influence on the estimated coordinates and tropospheric parameters. E.g. you could do more careful analysis and consider possible dependence on the on the localization of stations, antenna models, etc. (for example maybe for stations in high mountains or closer to equator some variant are better than others). This would improve the content of the manuscript.

We performed spatial and temporal analyses of all processed variants in order to assess the impact of different settings on tropospheric products. Zenith tropospheric delays from all variants were compared in such a way to enable assessing impact of any single processing change: 1) GO1-GO0 for mapping function and more precise a priori ZHD model, 2) GO2-GO1 and GO3-GO1 for different elevation angle cut-off, 3) GO4-GO1 for non-tidal atmospheric corrections, 4) GO5-GO4 for higher-order ionospheric corrections and, 5) GO6-GO4 for temporal resolution tropospheric horizontal gradients. Station-specific behavior is out of this paper and will be studied in future. New subsection (4.4) was added to the manuscript. However, we believe more detailed study on site-specific behaviour is out of the scope of this paper as it would require more time for analysis and additional space for text and figures. We will certainly use the dataset for it in future.

a) You wrote that the network was split into 10 sub-networks. In Figure 1 based on different colours I can distinguish only 6 clusters. It is better to change the markers and e.g. some clusters mark as squares.

The number of clusters was fixed (8 instead of 10) and Figure 1 was enhanced to distinguish station colours.

b) Did you use common stations to link the clusters in the network solution?

Nothing was fixed in the processing, while clustering and fiducial station definitions were dynamically adapted in the processing system.

c) How did you choose the clusters of stations? Based on the localization of the stations? I can see in Figure 1 that clusters are regional – stations which are located close to each other are in the same cluster, and the stations of each sub-networks are always the same. Is it an optimal solution of the sub-networks design?

Geographic clusters were only defined a priori while still possibly adapted dynamically within the processing. Number of used clusters, their size and selection of stations varied in different processing steps, e.g. sometimes using geographic clusters, sometimes random number of stations, sorted or conditioned for specific selections etc. The use of common station or other fixed cluster definition could lead to problems in combined solution when few or poor data at some station are available for linking the clusters. For linking clusters we always used baselines of maximum number of observations.

Santamaría-Gómez (2010) showed the results of processing of global network clustered into "dynamic subnetwork", where closer stations were distributed in different sub-networks in order to obtain a regular distribution based on station baseline geometry. They showed a noticeable improvement in the percentage of fixed ambiguities, especially before the year 2000, and also improvement of position repeatability and transformation parameters with respect to a "static sub-networks" solution. Did you test maybe this kind of clustering in your processing?
- Santamaría-Gómez, A. (2010), Estimation of crustal vertical movements with GPS in a geocentric frame, within the framework of the TIGA project, doctoral dissertation of the Observatoire de Paris.
- Santamaría-Gómez, A.; Bouin, M.-N.; Wöppelmann, G. (2009), Impact of subnetwork configuration on global scale GPS processing, EGU General Assembly 2009.

We haven't tested such a clustering, but our clustering approach is based on our long-term experience in developing daily and hourly (near real-time) processing in regional or global scope for estimating tropospheric parameters, coordinates or orbits. We suppose that suggested method might be of interest in a global network processing mainly for orbit determination, however, in a regional network such as EUREF Permanent Network, we prefer prioritizing shorter baselines for reducing the impact of orbit errors and for easier initial phase integer ambiguity resolution. We never use static sub-networks as it is risky and can not generally guarantee a high quality. We process clusters keeping a reasonable reference datum and other parameters for careful pre-elimination of unresolved ambiguities. The final IGS frame was then realized within the combined solution.

4. Section 4.1, lines 202-223: You wrote that you used an interactive procedure of validation of the fiducial stations. Can you be more specific on what this procedure was and how it works? Did you choose stations based on daily repeatability of their coordinates? What was your set of fiducial stations? IGS stations?

For validating a priori defined fiducial stations (IGS stations with precise IGS08 coordinates and velocities) we used the iterative procedure exploiting coordinate residuals at all active fiducial stations when applying Helmert transformation between the IGS08 coordinates and 7-day GOP combined solutions. The criteria for rejection of particular fiducial station were set 15, 15 and 30 mm for north, east and up components, respectively.

5. Section 4.2:
a) lines 233-253: It's a quite long paragraph about comparison of ZTD obtained from GOP Repro2 reprocessing to EUREF Repro1 products. We can expect that EUREF Repro 1 is worse than each version of Repro 2. The fact that some variant of the reprocessing is closer to EUREF Repro 1 does not mean that it is better. So, is it really useful to show such results? Does it bring any meaningful statistics? I think comparison to any external data (for instance ERA-Interim what is shown in next paragraph) is more interesting and conclusive.

We have removed this part suggested by other comments too. Thank you for suggestion.

b) lines 260-274, Figure 5: GNSS ZTD from each reprocessing compared to ZTD from ERA-Interim is characterized by a negative bias. We can also notice it in the EPN solution. Can you explain why the bias is negative?

Actually, we cannot explain it preciously. We expect the mean bias of about -1.8 mm (with uncertainty of 2-3 mm) is coming mainly from the ERA-Interim re-analysis. It means ERA-Interim is drier which could be related to the water vapour content underestimates. Such bias for the ERA-Interim has not been observed in a small dense network in Central Europe of the GNSS4SWEC Benchmark campaign in May-June 2013 (Dousa et al., 2016), but for the same dataset, a larger bias (- 4.9 mm) was observed for NCEP's Global Forecasting System. The mean bias -1.8 mm in Europe over 1996-2014 revealed to be rather stable (see Figure 6) and, it has been observed in all GNSS re-processing results in Europe (Pacione et al. 2017). Alternatively, the bias could be attributed to the numerical weather data processing method, however, within the GNSS4SWEC Benchmark campaign we processed ERA-Interim with two different software and methodology for calculating ZTD, compared them with two different GNSS processing methods while haven't found significant differences in results. The description was added (Section 4.2).

6. References, line 472: Please, change reference Pacione et al. (2017) to: Pacione, R., Araszkiewicz, A., Brockmann, E., and Dousa, J.: EPN Repro2: A reference GNSS tropospheric dataset over Europe, Atmos. Meas. Tech. Discuss., doi:10.5194/amt-2016-369, in review, 2016.

The reference has been corrected.

**Responses to Editor, Dr Olivier Bock.**

Dear authors, please find below a few additional comments to those provided by the two referees.

1. You write that the EUREF recommendations are followed which specify that "weekly coordinates should be used to estimate tropospheric parameters on a daily basis" (L129) and that the coordinates were fixed to these values (L144). Did you fix the coordinates for all stations or only the fiducial stations? Fixing station heights is known to produce biases in ZTD estimates due to un-modelled station motions (tidal and non-tidal, e.g. seasonal) and other error sources (because of the correlation between estimated parameters). Abrupt changes and drifts over time that impact stations height would then also map into ZTD estimates. Can you comment on the uncertainty in the ZTD estimates, and possibly also gradients, due your specific processing strategy?

We understand the point and your concerns. However, first we had no choice but follow the EUREF guidelines for tropospheric estimates as our primary goal was the contribution to EUREF. And second, it is also well known that weekly coordinates estimated are more accurate than daily estimates which particularly concerns of the height component and the most one correlated with ZTD parameters. It is also known that ZTDs are temporally correlated up to 1-2 days (Stoew and Elgered, 2005) suggesting to use a longer period than a single day for a proper decorrelation of coordinates and tropospheric parameters. Thus, in the last step of our procedure, the tropospheric parameters on daily basis were estimated with tightly constrained weekly coordinates (for all the stations). We believe that any drift over time is handled in this way while abrupt changes could be difficult to handle anyway using daily solutions. On a weekly basis, we could additionally apply quality control based on residuals from weekly combination for identifying and rejecting outliers on a daily basis. Generally, it would be difficult to assess the uncertainty of ZTD and gradient estimates due to our specific strategy as we cannot easily separate and evaluate errors propagated into tropospheric parameters due to un-modelled day-to-week station motions. We believe the uncertainty of our specific strategy is comparable to or lower than the method simultaneously estimating coordinates and tropospheric parameters. As it concerns to any other error sources, e.g. such as from precise products, we are not happy they still contaminate our solution, however, it would not be more beneficial to assimilate them into daily station coordinates which might be the case.

2. The relevance of this study is that several processing variants are produced with the same software. The results are thus not obscured by inter-software biases. However, the discussion of results from the different variants is quite short in the manuscript. The accuracy of tropospheric parameters is only analysed based on Table 5 and Figure 5 and 6. Table 5 compares the biases and standard deviations over all stations and all times for the different variants. It is striking that the differences in these numbers are tiny. I would not be surprised that a spatialized analysis reveals significant impact of changing the cut-off angle and mapping functions at sites in different climatic regions, in mountainous areas, or close to the sea. If relevant, I suggest that you complement the paper with spatialized results.

We performed spatial and temporal analyses of all processed variants in order to assess the impact of different settings on tropospheric products. Zenith tropospheric delays from all variants were compared in such a way to enable assessing impact of any single processing change: 1) GO1-GO0 for mapping function and more precise a priori ZHD model, 2) GO2-GO1 and GO3-GO1 for different elevation angle cut-off, 3) GO4-GO1 for non-tidal atmospheric corrections, 4) GO5-GO4 for higher-order ionospheric corrections and, 5) GO6-GO4 for temporal resolution tropospheric horizontal gradients. Station-specific behavior is out of this paper and will be studied in future. New subsection (4.4) was added to the manuscript. However, we believe more detailed study on site-specific behaviour

is out of the scope of this paper as it would require more time for analysis and additional space for text and figures. We will certainly use the dataset for it in future.

3. Table 4 comparing the GOP solutions to the outdated EUREF repro1 is not relevant. This comparison might be done as an initial consistency check of the new solutions compared to the legacy EUREF reference. I thus suggest removing this figure and the related text from the manuscript.

Table 4 and related text comparing GOP Repro2 with EUREF Repro1 were removed from the manuscript.

4. The temporal homogeneity of long time series is crucial when trends are to be estimated. Given that there is presently a high interest of the GNSS/climate community in estimating trends, I think it would be useful to complement the results with an analysis of trends for the different processing variants. There are many questions like: which cut-off angle and mapping functions choose to get the most homogenous time series? What is the impact of changing quality in GNSS observations over time? Again, the conclusions might be station dependent and both overall and spatialized analyses might be necessary to document them properly.

We added an analysis of trends using different processing variants. The analysis was limited to 12 stations with the longest data time-series. Trends ranged from -0.05 to 0.38 mm/year with formal errors of 0.01-0.02 mm/year. The most significant impact was observed due to the changing elevation angle cut-off reaching differences up to 1 mm/year in ZTD while the impact of any other strategy change was below 0.5 mm/year only. The manuscript was completed by Section 5.

5. The impact of the temporal resolution of gradient parameters is intriguing. Indeed, better accuracy is expected when combing the 6-hourly to 24-hourly estimates. With 4 times more observations the standard deviation is expected to be divided by a factor of 2. However, according to Table 5 the improvement is only by a factor of 1.3 suggesting there is serial correlation in the errors. Is this reduction factor is uniformly distributed over stations and stable over time? Can you be more specific about the correlation between gradients and other parameters suggested in the manuscript? (L280 and 309)

You are right, the real factor of the improvement when using additional observations is lower than the one theoretically expected indicating the correlations in the errors. The factor was found generally stable over all stations when ranging from 1.03 to 1.65 with the mean value of 1.35. The description and discussions was completed (Section 4.3).

6. The drift in the gradient estimates at station MALL (Fig. 7) is impressive. How did the ZTD estimates evolve during the period when gradients drifted? Did you detect other cases like this? In the case of MALL the cause was identified as a tracking problem. Did you detect other causes which could produce such drifts in gradients or ZTDs? It would be interesting to include a check on gradients as part of a data screening method. I suggest considering this idea in the discussion.

ZTDs at MALL stations were affected significantly too. During the same period, the period, also yearly mean ZTD differences to ERA-Interim steadily changed from about 3 mm to about -12 mm and immediately dropping down to -2 mm in 2008 after the antenna change. Short note added to the manuscript (Section 6).

Although the station MALL represented an extreme case, biases at other stations were observed too, e.g. GOPE (1996-2002), TRAB (1999-2008), CREU (2000-2002), HERS (1999-2001), GAIA (2008-2014)

and others. Site-specific, spatially or temporally correlated biases suggest different possible reasons such as site-instrumentation effects including the tracking quality and phase centre variation models, site-environment effects including multipath and seasonal variation (e.g. winter snow/ice coverage), edge-network effects when processing double-difference observations, spatially correlated effects in reference frame realization and possibly others. More detail investigation is out of the scope of this paper and will be studied in future. This short discussion added in manuscript (Section 6).

We fully agree that the assessing gradient parameters could be a valuable method as a part of ZTD data screening procedure. Short note added to the discussion (Section 6).

Other specific comments are given in the annotated PDF. Please also note the supplement to this comment
http://www.atmos-meas-tech-discuss.net/amt-2017-11/amt-2017-11-EC1-supplement.pdf

All specific comments were carefully resolved too.

**Supplementary materials**

These supplementary materials display geographical visualization of systematic errors (Figure 1) and standard deviations (Figure 2) from inter-comparisons of the GOP 2$^{nd}$ reprocessing variants. Zenith tropospheric delays were compared in the way to assess the impact of a 
[revised manuscript text omitted]

**3  Ensuring ZTD continuity at midnights**

When site tropospheric parameter time series generated from the 2[nd] EUREF reprocessing are applied to climate research, they should be free of artificial offsets in order to avoid misinterpretations (Bock et al., 2014). However, GNSS processing is commonly performed on a daily basis according to adopted standards for data and product dissemination. Thus far, EUREF analysis centres have provided independent daily solutions, although precise IGS products are combined and distributed on a weekly basis. Station coordinates are estimated on a daily basis and are later combined to form more stable weekly solutions. According to the EUREF analysis centre guidelines (http://www.epncb.oma.be/_documentation/guidelines/guidelines_analysis_centres.pdf), weekly coordinates should be used to estimate tropospheric parameters on a daily basis, but there are no requirements with which to guarantee the continuity of tropospheric parameters at midnights. Additionally, there are also discontinuities on a weekly basis, as neither daily coordinates nor hourly tropospheric parameters are combined across midnights between corresponding adjacent GPS weeks.

The impact of the 3-day combination was previously studied when assessing the tropospheric parameters stemming from the 2[nd] IGS reprocessing campaign 2016 (Dousa et al. 2016) in the GOP-TropDB (Győri and Douša, 2016). Figure 4 shows the hourly statistics when comparing two global tropospheric products from the analysis centre CODE (Centre of Orbit Determination in Europe) which differ in applying 1-day or 3-day combination within the final solution (Dach et al., 2014). The statistics is based on comparing 2-hour ZTD estimates from both solutions during 2013 while 1-sigma uncertainties over all stations are displayed as y-errorbars. The increased impact of 3-day solution on the ZTD accuracy can be observed close to midnights and indicates a 1-sigma uncertainty over differences in ZTDs at daily boundary stemming from 1-day and 3-day solutions. Actual differences in ZTDs are could be even 
[revised manuscript text omitted]
 tropospheric parameters, and particularly wet contribution, during the different seasons as it will be clear also in the next section.

**4.2 Zenith total delays**

We compared all reprocessed tropospheric parameters with respect to independent data from the ERA-Interim global reanalysis (Dee et al. 2011), which were developed and provided by the European Centre for Medium-Range Weather Forecasts (ECMWF) from 1969 to the present. For the period of 1996-2014, we calculated tropospheric parameters (namely ZTD and tropospheric horizontal linear gradients) from the NWM for all EPN stations using the GFZ (German Research Centre for Geosciences) ray-tracing software (Zus et al., 2014).

Besides ZTDs, Table 4 also summarizes comparisons of the tropospheric horizontal delays with those obtained from the ERA-Interim. It indicates a mean ZTD bias -1.8 mm for all comparisons (GNSS – NWM) which seems to be related to the ERA-Interim suggesting underestimates of the water vapour content. Similar bias has been observed for all other European GNSS re-processing products (Pacione et al., 2017). Alternatively, the bias could be attributed to the numerical weather data processing method. However, by processing ERA-Interim with two different software and methodologies within the GNSS4SWEC Benchmark campaign (Dousa et al., 2016) and by comparing them to two GNSS reference products based on different processing methods, we observed differences in bias bellow ±0.4 mm. On the other hand, no systematic errors were identified in the Benchmark campaign between ERA-Interim and two GNSS reference ZTD solutions when using a small dense network in Central Europe and a short period in May-June 2013. Large negative bias (-4.9 mm) was, however, observed for ZTD parameters derived from the NCEP's Global Forecasting System when compared to the same reference GNSS reference ZTD solutions.

[revised manuscript text omitted]

As in case of ZTD and coordinate assessment, tropospheric gradients also recorded the degradation when raising the elevation angle cut-off from 3 degrees to 7 degrees (GO2) or 10 degrees (GO3) and no impact was observed from additional modelling of high-order ionospheric effects (GO5), see Table 4. Mean standard deviations of the GO2 and GO3 solutions increased by 8% and 12%, respectively, which was visible over the whole period in monthly time series (not showed). No significant differences in temporal variations of mean biases of the north and east tropospheric gradients variants were identified while they shared a higher variability during the years 1996-2001.

Finally, comparing GO4 and GO6 solutions with ERA-Interim revealed that standard deviations dropped from 0.38 mm to 0.28 mm and from 0.40 mm to 0.29 mm for the east and north gradients, respectively. Worse performance of the GO4 solution is attributed to the fact that tropospheric horizontal gradients were estimated with a 6-hour sampling interval and a piece-wise linear function without the application of absolute or relative constraints. In such cases, increased correlations of these gradients with other parameters can cause additional instabilities in processing certain stations at specific times; these gradients can then absorb remaining errors in the GNSS analysis model. The mean biases of the tropospheric gradients are considered to be negligible, but we will demonstrate in the following section that some large systematic effects were indeed discovered and were attributed to the quality of GNSS signal tracking.

**4.4 Spatial and temporal ZTD analysis**

We performed spatial and temporal analyses of all processed variants in order to assess the impact of different settings on tropospheric products. Zenith tropospheric delays from all variants were compared in such a way to enable assessing impact of any single processing change: 1) GO1-GO0 for mapping function and more precise a priori ZHD model, 2) GO2-GO1 and GO3-GO1 for different elevation angle cut-off, 3) GO4-GO1 for non-tidal atmospheric corrections, 4) GO5-GO4 for higher-order ionospheric corrections and, 5) GO6-GO4 for temporal resolution tropospheric horizontal gradients. Station-specific behavior is out of this paper and will be studied in future.

Geographical maps of spatially distributed biases and standard deviations in ZTDs from all compared variants for the whole network are available within the supplementary materials. In the paper, we display only site-specific ZTD statistics with respect to the station ellipsoidal height, latitude and time in Figure 10, Figure 11 and Figure 12, respectively. Median, minimum and maximum values of station-wise total statistics are given in Table 5 demonstrating the impact of the higher-order effect is negligible as well as mean biases, but for the GO1-GO0 comparison. Generally, height dependences are supposed to be mainly due to higher magnitudes of ZTDs increasing the impact of individual models

and their uncertainties. The impact on standard deviations is dominant in the GO1 vs. GO0 comparison, while impacts on systematic errors are visible more or less in all comparisons, Figure 10.

Using actual mapping function and precise a priori ZHD from VMF1 instead of blind GMF/GPT models (GO1 vs. GO0), we observe negative systematic errors ranging from -1.52 to 0.70 mm and the median value -0.36 mm, according to Table 5, with a moderate latitudinal dependence, see Figure 11. A similar, but slightly larger negative bias of -0.94±0.28 mm was reported Kacmarik et al. (2017) studying 400 stations in the central Europe. Standard deviations in the table range from 0.69 mm to 3.82 mm, with a profound increase with latitude in Figure 11 suggesting the blind models perform worse at high latitudes. However, it is difficult to judge about the reason as it might be a product of mixed impact of a priori ZTD modelling, separating hydrostatic and wet component and applying mapping function. It suggests a more detailed study in future. Additionally, Figure 12 shows the effect grows with time which is attributed to the presence of more low-elevation observations as the elevation cut-off was updated gradually up to the horizon within the EUREF permanent network.

The impact of different elevation angle cut-off doesn't reveal any systematics in Figure 11. Biases for comparison of variants 3°/7° (GO2-GO1) and 3°/10° (GO3-GO1) range from -0.81 mm to 1.66 mm and -2.22 mm to 2.66 mm, respectively, and for standard deviations from 0.15 mm to 1.29 mm and 0.31 to 2.04 mm, see Table 5. As expected, the impact is larger for the GO3-GO1 differences and affected particularly some stations. Yearly biases exceeding ±2.5 mm were identified for BELL, DENT, MLVL, MOPS, POLV RAMO and SBG2 EPN stations (http://epncb.oma.be). Temporal dependences in the GO2-GO1 and GO3-GO1 comparisons, see Figure 12, show systematic errors growing together with increasing impact of low-elevation observations in time.

The impact of non-tidal atmospheric loading (GO4-GO1) seems to be strongly site-specific and doesn't reveal any latitudinal dependence in Figure 11. It however shows some degradation prior the year 2002, see Figure 12, which hasn't been understood yet. Biases and standard deviations in Table 5 range from -2.29 mm to 5.55 mm and from 0.68 mm to 4.72 mm, respectively. It represents one the largest impact in term of systematic errors and the second largest impact in term of standard deviations when compared to other comparison variants. Generally, the effect corresponds to the site-specific modelling of non-tidal atmospheric loading corrections and their partial compensations via blind pressure model (GPT) used at GO0 for individual stations. Standard deviations above 3 mm were observed at these stations: JOZE, MAD2, MADR, MDVO, MOPI, NYAL, SBG2, VENE and WETT.

The impact of higher-order ionospheric effect (GO5-GO4) is negligible at all stations demonstrating total statistics for all stations within ±0.3 mm with applying the y-range about 10 times smaller than in other panels in Figure 11. However, a strong latitudinal dependence is still visible in the figure and, a strong temporal variability shows yearly statistics up to ±0.4 mm in Figure 12. Both dependences are due to the changing magnitude of ionospheric corrections, increasing towards equator, and due to the solar magnetic activity cycles, reaching peaks around years 2001 and 2014.

The impact of stacking tropospheric gradients from 6-hour to daily estimates (GO6-GO4) is almost negligible for systematic errors which stay below ±1 mm. However, standard deviations range from 0.76 mm to 2.46 mm, growing towards lower latitudes, see Figure 11, which can be attributed to the increasing amount of water vapor content and its asymmetry imperfectly modelled by adding tropospheric gradients. Finally, there is no significant temporal variation observed in Figure 12.

**5    Impact of variants on long-term trend estimates**

We assessed the impact of processing variant settings on long-term trend estimates by analysing 12 EUREF stations providing the longest time-series of data. The trends were estimated using the least squares regression method applied on model

$$Y_t = \mu + \beta X_t + S_t + \varepsilon_t \tag{2}$$

where $\mu$ is the constant term of the model, $\beta X_t$ is the linear trend function with $\beta$ representing the trend magnitude, $S_t$ represents the seasonal term modelled by the sine wave function of time $X_t$ including seasonal, sub-seasonal and high-frequencies, and finally $\varepsilon_t$ is the noise in the data. Trend magnitudes were estimated using the original hourly ZTD estimates without any time-series homogenization, i.e. change-point detection and shift elimination. Data from all variants were processed for all selected stations and displayed in Figure 13. Trends ranged from -0.05 to 0.38 mm/year with formal errors of 0.01-0.02 mm/year. The most significant impact was observed due to the changing elevation angle cut-off reaching differences up to 1 mm/year in ZTD while the impact of any other strategy change was below 0.5 mm/year only.

**6    Tropospheric gradients biases vs quality of observations**

[revised manuscript text omitted]

---

## Editor Decision (ED1)

Dear authors,

Thank you for carefully revising the manuscript according to the comments from two referees and from myself. The manuscript has been significantly improved and augmented. I think it provides interesting results to the atmospheric science community, especially with the addenda of sections 4.4 and 5. Since I asked for these addenda, I didn't send the revised manuscript back to the referees but did the 2nd review myself. Please see the specific comments and questions below and a few edits in the annotated manuscript.

Specific comments

GNSS processing: did you include GPS and observations from other systems? I couldn't find this information.

P4L131-134 : I guess you refer to the monthly statistics of ZTD and/or gradient differences (GPS – ERAI) ? This is a post-processing QC, so maybe it should not be mentioned in this section.

P4L147-158 : on the impact of 3-day combination vs. 1-day solution. This discussion is useful and answers a comment from 1st referee. However, I think Figure 2 should not be presented in this paper since it is from a different analysis centre. Please removed the figure and revise the text.

P6L225-226: Explain why the improvement is smaller for GO4.

P7L255-258: Why do you think the residuals are dominated by errors due to modelling of tropospheric parameters, and particularly the wet contribution?

P7L266-278: this discussion calls for more careful interpretation. I think there is no evidence that the bias is due to ERA-Interim data (L268). If this statement is based on the GNSS comparison, assuming GNSS is the reference, it should be proved first that GNSS can sense the absolute ZTDs with an accuracy better than 1.8 mm. The fact that a similar bias was observed with other GNSS software doesn't mean that the bias is not in the GNSS estimates (L269). It just indicates that the bias is not software-dependent. A common bias might be due to the use of the same satellite products, similar tropospheric model, the regional-scale network…

P7L274-278: These comments don't add something to this study.

P8L299-301 and Figure 6-7: It is not clear what the statistics mean exactly. Are all values put together or is there an order of computation with respect to time and stations? It should be clearly explained how the means, standard deviations and error-bars are computed. The error bars seem very large for standard errors (by definition representing the uncertainty of the computed value). My interpretation is that the mean bias and standard of deviation of ZTD differences are first computed for each station and each month, and then mean and standard of deviation of these values are computed and the means are plotted +/- 1- standard of deviation. This would mean that the error bars represent the 1-sigma range (or dispersion) of values over the ensemble of stations. This is something different from standard error and uncertainty.

Table 4: similar comment: explain how the mean +/- dispersion (?) values are computed.

P9L333-342 and L350-359: recombine these two paragraphs

General comments on Section 4.4:

- The comparisons presented in this section are relevant and the discussion is well organised, comparison after comparison, referring to Table 5 and the different figures.
- I suggest to go in the same order for each discussion, commenting first on the median and min/max values in Table 5, and then on the latitudinal and height variations, and finally on the time evolution.
- Since latitudinal variations are more relevant I suggest switching the order of Figure 8 and 9.
- There is in general no trend with height, except in GO1-GO0. However, the increased scatter at low heights might be mentioned (for all comparisons). Is there an explanation?
- Add median values (e.g. as solid line) on the plots in Figure 10. It is important to check if there are drifts over time.
- The maps provided in the supplementary material could be included in the manuscript and referred to in complement to the latitudinal and height variations (Fig. 8, 9).
- Instead, it would be appropriate to provide in the supplementary material a table or a text file with the results plotted in the maps and in Figure 9 (overall statistics by station, including latitude, longitude and station height).
- Use the term 'bias' throughout the text to be consistent with the figures and tables, instead of using sometimes 'systematic errors'.

Specific comments on Section 4.4

P10L383: I think the differential performance of GMF/GPT and VMF1 has been studied and published by Boehm et al., and the increased errors of GMF/GPT at higher latitudes is a known issue. Please check the literature and provide adequate comments and references on this issue.

P10L386: Effect is attributed to more low-elevation observations over time. It would be nice that you document the time evolution of the number of (low-elevation) observations to provide evidence for this effect. Since you processed the data you have all the necessary information. This is usually not possible to check for end users (e.g. from the meteorological community).

L395: conclude on the sensitivity of results on the cutoff angle.

L398: could the degradation prior to 2002 be due to a change in the processing method or in pressure data used to compute the atmospheric loading?

L401-403: the scatter in bias and std is the largest for this comparison (GO4-GO1). Since coordinates are fixed, I guess that uncorrected atmospheric loading in GO1 solution is compensated by ZTD biases. Is this what you mean?

L414: why would the asymmetry be more imperfectly modelled with one tropospheric gradient every 6 hours rather than one per 24 hours? It seems more logical that the model with more parameters is better.

General comments on Section 5

The goal here is to assess the impact of variants on trend estimates. Though a detailed analysis of trend estimates is out of scope of this paper, the results shown in Figure 11 should be more thoroughly commented and maybe completed with mean values per variant, or differences of variants like in section 4.4. It seems to me that GO0 and GO1 yield very similar trends, which means that the mapping function and ZHD don't have a strong impact. Cutoff angle has an impact (as previously shown by Ning and Elgered, 2012). Variants GO4, GO5, and GO6 are very similar, but not consistent with GO1, which means that non-tidal atmospheric loading has a significant impact.

It would be very valuable if trends were computed for all stations and all variants and the mean values and std summarized in a Table per variant. Conclusions could thus be more statistically significant. Then I would also suggest to provide the trend and uncertainty estimates per station as a supplement for further intercomparison with similar publications in the COST Special Issue (Baldisz et al., AMT-9-4861-2016, Klos et al. AMT-2016-385).

Specific comments on Section 5

P11L419 : add a reference to Eq (2), e.g. Weatherhead et al., 1998 ; Bock et al., 2014

Weatherhead, E. C., et al. (1998), Factors affecting the detection of trends: Statistical considerations and applications to environmental data, J. Geophys. Res., 103(D14), 17,149–17,161, doi:10.1029/98JD00995.

P11L423 : seasonal, sub-seasonal and high-frequencies => be more specific: e.g., annual and $2^{nd}$, 3rd… harmonics.

P11L427 : Note that the noise is assumed white and thus the formal errors of the trend estimates are underestimated by a factor 2-4 (Nilsson and Elgered, 2008 ; Klos et al. AMT-2016-385).

Nilsson, T., and G. Elgered (2008), Long-term trends in the atmospheric water vapor content estimated from ground-based GPS data, J. Geophys. Res., 113, D19101, doi:10.1029/2008JD010110.

P11L428-429 : The impact of cutoff angle was already reported by Ning and Elgered (2012).

Ning, T., and G. Elgered (2012), Trends in the atmospheric water vapor content from ground-based GPS: The impact of the elevation cutoff angle, IEEE J. Sel. Top. Appl. Earth Obs. Remote Sens., 5, 744–751, doi:10.1109/JSTARS.2012.2191392.

P11L428-429 : differences don't reach 1 or 0.5 mm/year, please correct.

Section 6 : suggest to move it after section 3 (so change numbering of subsequent sections) as the focus is more on observation level and data processing.

P11L431-432 : is this web interface available to the scientific community ?

P11L454 : could you add the ZTD difference series on Figure 12 ?

At the end of section 6 it is not clear if the problematic stations/periods were removed from the dataset analysed in previous sections?

Table 4: 0.43 is repeated in the last column

[revised manuscript text omitted]

also applied iterative procedure for excluding residuals exceeding 3-sigma of standard deviation
calculated from the compared differences (Gyori and Dousa, 2016).

**3   Ensuring ZTD continuity at midnights**

When site tropospheric parameter time series generated from the 2nd EUREF reprocessing are applied
to climate research, they should be free of artificial offsets in order to avoid misinterpretations (Bock
et al., 2014). However, GNSS processing is commonly performed on a daily basis according to adopted
standards for data and product dissemination. Thus far, EUREF analysis centres have provided
independent daily solutions, although precise IGS products are combined and distributed on a weekly
basis. Station coordinates are estimated on a daily basis and are later combined to form more stable
weekly solutions. According to the EUREF analysis centre guidelines
(http://www.epncb.oma.be/_documentation/guidelines/guidelines_analysis_centres.pdf), weekly
coordinates should be used to estimate tropospheric parameters on a daily basis, but there are no
requirements with which to guarantee the continuity of tropospheric parameters at midnights.
Additionally, there are also discontinuities on a weekly basis, as neither daily coordinates nor hourly
tropospheric parameters are combined across midnights between corresponding adjacent GPS weeks.

The impact of the 3-day combination was previously studied when assessing the tropospheric
parameters stemming from the 2nd IGS reprocessing campaign 2016 (Dousa et al. 2016) in the GOP-
TropDB (Győri and Douša, 2016). Figure 2 shows the hourly statistics when comparing two global
tropospheric products from the analysis centre CODE (Centre of Orbit Determination in Europe) which
differ in applying 1-day or 3-day combination within the final solution (Dach et al., 2014). The statistics
is based on comparing 2-hour ZTD estimates from both solutions during 2013 while 1-sigma
uncertainties over all stations are displayed as y-errorbars. The increased impact of 3-day solution on
the ZTD accuracy can be observed close to midnights and indicates a 1-sigma uncertainty over
differences in ZTDs at daily boundary stemming from 1-day and 3-day solutions. Actual differences in

[revised manuscript text omitted]
 tropospheric parameters, and particularly wet contribution, during the different seasons as it will be clear also in the next section.

**4.2 Zenith total delays**

We compared all reprocessed tropospheric parameters with respect to independent data from the ERA-Interim global reanalysis (Dee et al. 2011), which were developed and provided by the European Centre for Medium-Range Weather Forecasts (ECMWF) from 1969 to the present. For the period of 1996-2014, we calculated tropospheric parameters (namely ZTD and tropospheric horizontal linear gradients) from the NWM for all EPN stations using the GFZ (German Research Centre for Geosciences) ray-tracing software (Zus et al., 2014).

Besides ZTDs, Table 4 also summarizes comparisons of the tropospheric horizontal delays with those obtained from the ERA-Interim. It indicates a mean ZTD bias -1.8 mm for all comparisons (GNSS – NWM) which seems to be related to the ERA-Interim suggesting underestimates of the water vapour content. Similar bias has been observed for all other European GNSS re-processing products (Pacione et al., 2017). Alternatively, the bias could be attributed to the numerical weather data processing method. However, by processing ERA-Interim with two different software and methodologies within the GNSS4SWEC Benchmark campaign (Dousa et al., 2016) and by comparing them to two GNSS reference products based on different processing methods, we observed differences in bias bellow ±0.4 mm. On the other hand, no systematic errors were identified in the Benchmark campaign between ERA-Interim and two GNSS reference ZTD solutions when using a small dense network in Central Europe and a short period in May-June 2013. Large negative bias (-4.9 mm) was, however, observed for ZTD parameters derived from the NCEP's Global Forecasting System when compared to the same reference GNSS reference ZTD solutions.

[revised manuscript text omitted]

As in case of ZTD and coordinate assessment, tropospheric gradients also recorded the degradation
when raising the elevation angle cut-off from 3 degrees to 7 degrees (GO2) or 10 degrees (GO3) and
no impact was observed from additional modelling of high-order ionospheric effects (GO5), see Table
4. Mean standard deviations of the GO2 and GO3 solutions increased by 8% and 12%, respectively,
which was visible over the whole period in monthly time series (not  No significant differences
in temporal variations of mean biases of the north and east tropospheric gradients variants were
identified while they shared a higher variability during the years 1996-2001.

Finally, comparing GO4 and GO6 solutions with ERA-Interim revealed that standard deviations dropped
from 0.38 mm to 0.28 mm and from 0.40 mm to 0.29 mm for the east and north gradients, respectively.
Worse performance of the GO4 solution is attributed to the fact that tropospheric horizontal gradients
were estimated with a 6-hour sampling interval and a piece-wise linear function without the
application of absolute or relative constraints. In such cases, increased correlations of these gradients
with other parameters can cause additional instabilities in processing certain stations at specific times;
these gradients can then absorb remaining errors in the GNSS analysis model. The mean biases of the
tropospheric gradients are considered to be negligible, but we will demonstrate in the following
section that some large systematic effects were indeed discovered and were attributed to the quality
of GNSS signal tracking.

**4.4  Spatial and temporal ZTD analysis**

We performed spatial and temporal analyses of all processed variants in order to assess the impact of
different settings on tropospheric products. Zenith tropospheric delays from all variants were
compared in such a way to enable assessing impact of any single processing change: 1) GO1-GO0 for
mapping function and more precise a priori ZHD model, 2) GO2-GO1 and GO3-GO1 for different
elevation angle cut-off, 3) GO4-GO1 for non-tidal atmospheric corrections, 4) GO5-GO4 for higher-
order ionospheric corrections and, 5) GO6-GO4 for temporal resolution tropospheric horizontal
gradients. Station-specific behavior  will be studied in future.

Geographical maps of spatially distributed biases and standard deviations in ZTDs from all compared
variants for the whole network are available within the supplementary materials. In the paper, we
display only site-specific ZTD statistics with respect to the station ellipsoidal height, latitude and time
in Figure 8, Figure 9 and Figure 10, respectively. Median, minimum and maximum values of station-
wise total statistics are given in Table 5

Using actual mapping function and precise a priori ZHD from VMF1 instead of blind GMF/GPT models
(GO1 vs. GO0), we observe negative systematic errors ranging from -1.52 to 0.70 mm and the median
value -0.36 mm, according to Table 5, with a moderate latitudinal dependence, see Figure 9.
Standard deviations in the table range from 0.69 mm to 3.82 mm, with
a profound increase with latitude in Figure 9 suggesting the blind models perform worse at high
latitudes. However, it is difficult to judge about the reason as it might be a product of mixed impact of
a priori ZTD modelling, separating hydrostatic and wet component and applying mapping function. It
suggests a more detailed study in future. Additionally, Figure 10 shows the effect grows with time
which is attributed to the presence of more low-elevation observations as the elevation cut-off was
updated gradually up to the horizon within the EUREF permanent network.

The impact of different elevation angle cut-off doesn't reveal any systematics in Figure 9. Biases for
comparison of variants 3°/7° (GO2-GO1) and 3°/10° (GO3-GO1) range from -0.81 mm to 1.66 mm and
-2.22 mm to 2.66 mm, respectively, and for standard deviations from 0.15 mm to 1.29 mm and 0.31
to 2.04 mm, see Table 5. As expected, the impact is larger for the GO3-GO1 differences and affected
particularly some stations. Yearly biases exceeding ±2.5 mm were identified for BELL, DENT, MLVL,
MOPS, POLV RAMO and SBG2 EPN stations (http://epncb.oma.be). Temporal dependences in the GO2-
GO1 and GO3-GO1 comparisons, see Figure 10, show systematic errors growing together with
increasing impact of low-elevation observations in time.

The impact of non-tidal atmospheric loading (GO4-GO1) seems to be strongly site-specific and doesn't
reveal any latitudinal dependence in Figure 9. It however shows some degradation prior the year 2002,
see Figure 10, which hasn't been understood yet. Biases and standard deviations in Table 5 range from
-2.29 mm to 5.55 mm and from 0.68 mm to 4.72 mm, respectively. It represents one the largest impact
in term of systematic errors and the second largest impact in term of standard deviations when
compared to other comparison variants. Generally, the effect corresponds to the site-specific
modelling of non-tidal atmospheric loading corrections and their partial compensations via blind
pressure model (GPT) used  for individual stations. Standard deviations above 3 mm were
observed at these stations: JOZE, MAD2, MADR, MDVO, MOPI, NYAL, SBG2, VENE and WETT.

The impact of higher-order ionospheric effect (GO5-GO4) is negligible at all stations demonstrating
total statistics for all stations within ±0.3 mm with applying the y-range about 10 times smaller than in
other panels in Figure 9. However, a strong latitudinal dependence is still visible in the figure and, a
strong temporal variability shows yearly statistics up to ±0.4 mm in Figure 10. Both dependences are
due to the changing magnitude of ionospheric corrections, increasing towards equator, and due to the
solar magnetic activity cycles, reaching peaks around years 2001 and 2014.

The impact of stacking tropospheric gradients from 6-hour to daily estimates (GO6-GO4) is almost
negligible for systematic errors which stay below ±1 mm. However, standard deviations range from
0.76 mm to 2.46 mm, growing towards lower latitudes, see Figure 9, which can be attributed to the
increasing amount of water vapor content and its asymmetry imperfectly modelled by adding
tropospheric gradients. Finally, there is no significant temporal variation observed in Figure 10.

**5 Impact of variants on long-term trend estimates**

We assessed the impact of processing variant settings on long-term trend estimates by analysing 12 EUREF stations providing the longest time-series of data. The trends were estimated using the least squares regression method applied on model

$$Y_t = \mu + \beta X_t + S_t + \varepsilon_t \qquad (2)$$

where $\mu$ is the constant term of the model, $\beta X_t$ is the linear trend function with $\beta$ representing the trend magnitude, $S_t$ represents the seasonal term modelled by the sine wave function of time $X_t$ including seasonal, sub-seasonal and high-frequencies, and finally $\varepsilon_t$ is the noise in the data. Trend magnitudes were estimated using the original hourly ZTD estimates without any time-series homogenization, i.e. change-point detection and shift elimination. Data from all variants were processed for all selected stations and displayed in Figure 11. Trends ranged from -0.05 to 0.38 mm/year with formal errors of 0.01-0.02 mm/year. The most significant impact was observed due to the changing elevation angle cut-off reaching differences up to 1 mm/year in ZTD while the impact of any other strategy change was below 0.5 mm/year only.

**6 Tropospheric gradients biases vs quality of observations**

[revised manuscript text omitted]

(a)

(c)

**Figure 3: Charts of 4 variations on representations of tropospheric parameters. Right (b), (d) and left (a), (c) panels**
**display estimates made with and without midnight combinations, respectively. Top (a), (b) and bottom (c), (d) panels**
**display the piecewise constant and the linear model, respectively.**

[Figure]

(a)                                               (b)

(c)                                               (d)

**Figure 4: Four variations in representation of tropospheric parameters. Right (b), (d) and left (a), (c) panels display**
**estimates with and without midnight combinations, respectively. Top (a), (b) and bottom (c), (d) panels display the**
**piecewise constant and the piecewise linear model, respectively.**

[Figure]

**Figure 5: Statistics of the daily reference system realization: a) RMS of residuals at fiducial stations (representing the**
**total, height and position); b) number of stations (all and accepted after an iterative control)**

[Figure]

**Figure 6: Monthly means of bias and standard deviation of official GOP ZTD product compared to those of the ERA-Interim.**
**Error bars indicate standard errors of mean values over all compared stations.**

[Figure]

**Figure 7: Monthly means of bias and standard deviation of tropospheric horizontal north (N-GRD) and east (E-GRD)**
**gradients compared to those obtained by ERA-Interim. Note: Similar products are almost superposed. Error bars indicate**
**standard errors of mean values over all compared stations plotted from the zero y-axis to emphasise seasonal variations**
**and trends. Error bars are displayed for north gradients only, however, being representative for the east gradients too.**

[Figure]

**Figure 8: Dependence of ZTD systematic errors (blue) and standard deviations (red) from inter-comparisons of GOP 2nd**
**reprocessing solution variants on station ellipsoidal height. Note different y-range for the GO5 vs. GO4 comparison.**

[Figure]

**Figure 9: Dependence of ZTD systematic errors (blue) and standard deviations (red) from inter-comparisons of GOP 2[nd]**
**reprocessing solution variants on station latitude. Note different y-range for the GO5 vs. GO4 comparison.**

[Figure]

**Figure 10:** Dependence of ZTD systematic errors (blue) and standard deviations (red) from inter-comparisons of GOP 2nd
reprocessing solution variants on year. Note different y-range for the GO5 vs. GO4 comparison.

[Figure]

**Figure 11: Long-term ZTD trend estimates and their formal errors (error bars) for all processing variants.**

[Figure]

**Figure 12: MALL station - monthly mean differences in tropospheric horizontal gradients with respect to the ERA-Interim.**

[Figure]

**Figure 13: Low-elevation tracking problems at the MALL station during the period of 2003-2008. From left-top to right-**
**bottom: January 2002, 2004, 2006 and 2008 (courtesy of the EPN Central Bureau, ROB).**

[Figure]

**Figure 14: Sky plots before (left) and after (right) replacing the malfunctioning antenna at the MALL site (Oct 30, 2008).**
**Black dots indicates single-frequency observations available only.**

---

## Author Response (AR2)

Responses to Editor, Dr Olivier Bock.

All corresponding major revisions are marked in manuscript using red colour.

Dear authors,
Thank you for carefully revising the manuscript according to the comments from two referees and from myself. The manuscript has been significantly improved and augmented. I think it provides interesting results to the atmospheric science community, especially with the addenda of sections 4.4 and 5. Since I asked for these addenda, I didn't send the revised manuscript back to the referees but did the 2nd review myself. Please see the specific comments and questions below and a few edits in the annotated manuscript.

Specific comments
GNSS processing: did you include GPS and observations from other systems? I couldn't find this information.

GPS data were used only as explained at P3L103 and indicated in Table 1.

P4L131-134: I guess you refer to the monthly statistics of ZTD and/or gradient differences (GPS – ERAI)? This is a post-processing QC, so maybe it should not be mentioned in this section.

The corresponding sentences moved to Section 5.3 (originally 4.2).

P4L147-158: on the impact of 3-day combination vs. 1-day solution. This discussion is useful and answers a comment from 1st referee. However, I think Figure 2 should not be presented in this paper since it is from a different analysis centre. Please removed the figure and revise the text.

The figure was removed and the paragraph revised accordingly.

P6L225-226: Explain why the improvement is smaller for GO4.

It is not straightforward to compare the two results of different processing and different station set. As the effect depends on selected stations, a slightly higher impact in a global scale might be attributed to the distribution of stations including differences in term of latitude and altitude.

P7L255-258: Why do you think the residuals are dominated by errors due to modelling of tropospheric parameters, and particularly the wet contribution?

The quality of the troposphere modelling in GNSS analysis has a high seasonal dependence (in Europe at least) which is clearly visible from the comparison of tropospheric products either for GNSS vs. NWM or GNSS vs. GNSS, reaching up to a factor of 2-3 higher standard deviations in summer for both zenith total delays and tropospheric gradients. Obviously, the GNSS tropospheric model is simplified in term of modelling a horizontal tropospheric asymmetry as well as dependence on elevation angle. The former is mainly the effect due to a high spatio-temporal variability of humidity in the troposphere. The tropospheric delay has definitely the most pronounced seasonal signal in GNSS analysis and resulting ZTDs are correlated with estimated station height. It is well-known that the modelling of the first-order tropospheric asymmetry improves station horizontal position, however, the effect is about an order smaller compared to ZTD variation and its impact on height. We tried to reformulate the sentence considering that not all readers are familiar with GNSS analysis.

P7L266-278: this discussion calls for more careful interpretation. I think there is no evidence that the bias is due to ERA-Interim data (L268). If this statement is based on the GNSS comparison, assuming GNSS is the reference, it should be proved first that GNSS can sense the absolute ZTDs with an accuracy better than 1.8 mm. The fact that a similar bias was observed with other GNSS software doesn't mean that the bias is not in the GNSS estimates (L269). It just indicates that the bias is not software-dependent. A common bias might be due to the use of the same satellite products, similar tropospheric model, the regional-scale network…

Thank you for your comment, the paragraph was revised.

P7L274-278: These comments don't add something to this study.

The comments were removed.

P8L299-301 and Figure 6-7: It is not clear what the statistics mean exactly. Are all values put together or is there an order of computation with respect to time and stations? It should be clearly explained how the means, standard deviations and error-bars are computed. The error bars seem very large for standard errors (by definition representing the uncertainty of the computed value). My interpretation is that the mean bias and standard of deviation of ZTD differences are first computed for each station and each month, and then mean and standard of deviation of these values are computed and the means are plotted +/- 1- standard of deviation. This would mean that the error bars represent the 1-sigma range (or dispersion) of values over the ensemble of stations. This is something different from standard error and uncertainty.

The correspoding paragraph and figure were corrected using the term 1-sigma range of values over the ensemble of stations.

Table 4: similar comment: explain how the mean +/- dispersion (?) values are computed.

The desription was added in Section 5.3 (originally 4.2), the second paragraph.

P9L333-342 and L350-359: recombine these two paragraphs

Originally, the two paragraphs were related to different description – Table 4 vs Figure 7. Paragraphs in Section 5.3 (originally 4.2) were revised starting with the description of overall statistics in Table 4 (and all variants) followed by description of impact of gradient combination.

General comments on Section 4.4:
- The comparisons presented in this section are relevant and the discussion is well organised, comparison after comparison, referring to Table 5 and the different figures.
- I suggest to go in the same order for each discussion, commenting first on the median and min/max values in Table 5, and then on the latitudinal and height variations, and finally on the time evolution.
- Since latitudinal variations are more relevant I suggest switching the order of Figure 8 and 9.

The Section 5.5 (originally 4.4) was revised according to the abovementioned suggestions and the figures were also changed. The geographical maps provided in supplements were added and commented.

- There is in general no trend with height, except in GO1-GO0. However, the increased scatter at low heights might be mentioned (for all comparisons). Is there an explanation?

The explanation is supposed to be mainly due to the general increase of ZTD (both ZHD and ZWD) with the decrease of altitude, consequently increasing an overall impact of differences in applied models.

- Add median values (e.g. as solid line) on the plots in Figure 10. It is important to check if there are drifts over time.

Yearly mean biases and standard deviations were included to Figure 14 (originally 10).

- The maps provided in the supplementary material could be included in the manuscript and referred to in complement to the latitudinal and height variations (Fig. 8, 9).

The maps provided in supplements were added and referenced in Section 5.5 (originally 4.4).

- Instead, it would be appropriate to provide in the supplementary material a table or a text file with the results plotted in the maps and in Figure 9 (overall statistics by station, including latitude, longitude and station height).

ZTD trends for all variants estimated for 172 EPN stations with time series over 10 years were provided in table as supplements additionally indicating lengths of time series, number of available data and mean formal errors of the trend estimates.

- Use the term 'bias' throughout the text to be consistent with the figures and tables, instead of using sometimes 'systematic errors'. Specific comments on Section 4.4

Corrected.

P10L383: I think the differential performance of GMF/GPT and VMF1 has been studied and published by Boehm et al., and the increased errors of GMF/GPT at higher latitudes is a known issue. Please check the literature and provide adequate comments and references on this issue.

Thank you. We included the reference in the discussion. Our results are consistent with findings by Steinberger at al. (2009). Our results demonstrate a partial compensation of the atmospheric loading effect by using a blind model compared to a model based on actual weather conditions. In case the atmospheric loading effect was not corrected for, the errors are assimilated mainly to zenith total delays (less in the former

P10L386: Effect is attributed to more low-elevation observations over time. It would be nice that you document the time evolution of the number of (low-elevation) observations to provide evidence for this effect. Since you processed the data you have all the necessary information. This is usually not possible to check for end users (e.g. from the meteorological community).

New figure – example for WTZR station – was added indicating an increase in number of observations at low elevation angles during 1996-2015, which is, generally, common to a majority of EPN stations.

L395: conclude on the sensitivity of results on the cutoff angle.

Concluding sentence completed.

L398: could the degradation prior to 2002 be due to a change in the processing method or in pressure data used to compute the atmospheric loading?

The strategy was the same during all the 2nd reprocessing. As the degradation is observed only for G04-G01 comparison, it might be related to the quality of pressure data used to compute atmospheric loading.

L401-403: the scatter in bias and std is the largest for this comparison (GO4-GO1). Since coordinates are fixed, I guess that uncorrected atmospheric loading in GO1 solution is compensated by ZTD biases. Is this what you mean?

Yes, the sentences were really confusing and we modified them to be clear now, hopefully.

L414: why would the asymmetry be more imperfectly modelled with one tropospheric gradient every 6 hours rather than one per 24 hours? It seems more logical that the model with more parameters is better.

The plot shows only standard deviations from the comparison of GO4 vs GO6, i.e. 6-h vs. 24-h gradient model, thus it does not suggest which parameterization is better. The 6-h piecewise liner model for gradients could be helpful for more precise modelling of temporal variability in the troposphere. On the other hand, none of the parameterization is able to capture a higher-order asymmetry which is increasing towards the equator as mainly caused by humidity in the atmosphere. We modified the sentence to be clear.

General comments on Section 5
The goal here is to assess the impact of variants on trend estimates. Though a detailed analysis of trend estimates is out of scope of this paper, the results shown in Figure 11 should be more thoroughly commented and maybe completed with mean values per variant, or differences of variants like in section 4.4. It seems to me that GO0 and GO1 yield very similar trends, which means that the mapping function and ZHD don't have a strong impact. Cutoff angle has an impact (as previously shown by Ning and Elgered, 2012). Variants GO4, GO5, and GO6 are very similar, but not consistent with GO1, which means that non-tidal atmospheric loading has a significant impact. It would be very valuable if trends were computed for all stations and all variants and the mean values and std summarized in a Table per variant. Conclusions could thus be more statistically significant. Then I would also suggest to provide the trend and uncertainty estimates per station as a supplement for further intercomparison with similar publications in the COST Special Issue (Baldysz et al., AMT-9-4861-2016, Klos et al. AMT-2016-385).

In total, 172 EPN stations were finally selected according to time-series span over 10 years at least. For all these stations, the trend analysis was performed without data homogenization or outlier rejection as our focus was purely on impact of GOP variants. Site-by-site estimated trends from all variants are provided in supplementary materials completed by time-span information, number of data and mean trend formal errors. Description in new Section 6 (originally 5) was revised according to the above suggestions.

Specific comments on Section 5
P11L419: add a reference to Eq (2), e.g. Weatherhead et al., 1998 ; Bock et al., 2014
*Weatherhead, E. C., et al. (1998), Factors affecting the detection of trends: Statistical considerations and applications to environmental data, J. Geophys. Res., 103(D14), 17,149–17,161, doi:10.1029/98JD00995.*

References added. The second reference alredy used at other place.

*P11L423: seasonal, sub-seasonal and high-frequencies => be more specific: e.g., annual and $2_{nd}$, 3rd… harmonics.*

Corrected:  including annual, $2^{nd}$ harmonics (semi-annual) and daily.

P11L427: Note that the noise is assumed white and thus the formal errors of the trend estimates are underestimated by a factor 2-4 (Nilsson and Elgered, 2008 ; Klos et al. AMT-2016-385).
*Nilsson, T., and G. Elgered (2008), Long-term trends in the atmospheric water vapor content estimated from ground-based GPS data, J. Geophys. Res., 113, D19101, doi:10.1029/2008JD010110.*

Thank you. Reference added.

P11L428-429: The impact of cutoff angle was already reported by Ning and Elgered (2012).
*Ning, T., and G. Elgered (2012), Trends in the atmospheric water vapor content from ground-based GPS: The impact of the elevation cutoff angle, IEEE J. Sel. Top. Appl. Earth Obs. Remote Sens., 5, 744–751, doi:10.1109/JSTARS.2012.2191392.*

Reference added, used here and in also in Section 5.3 (originally 4.4).

P11L428-429: differences don't reach 1 or 0.5 mm/year, please correct.

Corrected.

Section 6: suggest to move it after section 3 (so change numbering of subsequent sections) as the focus is more on observation level and data processing.

Moved.

P11L431-432: is this web interface available to the scientific community?

The actual address is temporary, but it is being prepared for IGS Tropospheric WG web-pages www.igs.org.

P11L454: could you add the ZTD difference series on Figure 12? At the end of section 6 it is not clear if the problematic stations/periods were removed from the dataset analysed in previous sections?

Monthly ZTD biases added in the figure. The problematic stations/periods mentioned at the end of Section 4 (originally 6) were not removed from the dataset analysed in Section 6 as there was no scope to set objective criteria for definition of problematic stations. This should be a part of follow-on study.

Table 4: 0.43 is repeated in the last column

All values 0.43 in the last column of the table were corrected. Thank you for this notice.

---

## Author Response (AR3)

Responses to Dr Olivier Bock on

**Tropospheric products of the 2nd European GNSS reprocessing (1996-2014)**
by Jan Dousa et al.

Dear authors,

Thank you for implementing all the major changes and additions suggested in my previous review.
I think this work now brings significant answers to questions about the impact of various processing options on tropospheric parameter estimation, including ZTDs, gradients, and ZTD trends.
The results are well documented and discussed in the manuscript.
I send you back the last version of the manuscript with a few corrections to clarify some of the statements.
Please note that in my previous review, there was also an annotated manuscript attached. I think you didn't consider it as almost all corrections/edit were unapplied. I repeated some of them in the new version, but urge you to re-read the previous one as well.
Best regards,
Olivier.

All corresponding revisions are marked in the manuscript using the red colour. Comments from both versions were considered now.  We apology for unapplied corrections from the second part (annotated manuscript) of your previous review which was an unnoticed mistake.

Best regards

Jan Dousa, Pavel Vaclavovic, Michal Elias